# Archaic introgression contributed to shape the adaptive modulation of angiogenesis and cardiovascular traits in human high-altitude populations from the Himalayas

**Giulia Ferraretti**[1†], **Paolo Abondio**[2†], **Marta Alberti**[1], **Agnese Dezi**[3], **Phurba T Sherpa**[4], **Paolo Cocco**[5], **Massimiliano Tiriticco**[5], **Marco Di Marcello**[5], **Guido Alberto Gnecchi-Ruscone**[6], **Luca Natali**[5,7], **Angela Corcelli**[8], **Giorgio Marinelli**[5], **Davide Peluzzi**[5], **Stefania Sarno**[1†], **Marco Sazzini**[1,9*†]

[1]Laboratory of Molecular Anthropology and Centre for Genome Biology, Department of Biological, Geological and Environmental Sciences, University of Bologna, Bologna, Italy; [2]Department of Cultural Heritage, Ravenna Campus, University of Bologna, Bologna, Italy; [3]Department of Emergency and Organ Transplantation, University of Bari Aldo Moro, Bari Aldo Moro, Italy; [4]Mount Everest Summiters Club, Rolwaling, Dolakha, Nepal; [5]Explora Nunaat International, Montorio al Vomano, Teramo, Italy; [6]Department of Archaeogenetics, Max Planck Institute for Evolutionary Anthropology, Leipzig, Germany; [7]Italian Institute of Human Paleontology, Rome, Italy; [8]Department of Basic Medical Science, Neuroscience and Sense Organs, University of Bari Aldo Moro, Bari, Italy; [9]Interdepartmental Centre Alma Mater Research Institute on Global Changes and Climate Change, University of Bologna, Bologna, Italy

**\*For correspondence:**
marco.sazzini2@unibo.it

[†]These authors contributed equally to this work

**Competing interest:** The authors declare that no competing interests exist.

## eLife Assessment

This study presents **valuable** findings on what networks of genes were impacted by introgression from Denisovans, to identify the biological functions involved in high-altitude adaptation in Tibet. This study applies **solid** and previously validated methodology to identify genes with signatures of both introgression and positive selection. This paper would be of interest to population geneticists, anthropologists, and scientists studying the genetic basis underlying high-altitude adaptation.

**Abstract** It is well established that several *Homo sapiens* populations experienced admixture with extinct human species during their evolutionary history. Sometimes, such a gene flow could have played a role in modulating their capability to cope with a variety of selective pressures, thus resulting in archaic adaptive introgression events. A paradigmatic example of this evolutionary mechanism is offered by the *EPAS1* gene, whose most frequent haplotype in Himalayan high-landers was proved to reduce their susceptibility to chronic mountain sickness and to be introduced in the gene pool of their ancestors by admixture with Denisovans. In this study, we aimed at further expanding the investigation of the impact of archaic introgression on more complex adaptive responses to hypobaric hypoxia evolved by populations of Tibetan/Sherpa ancestry, which have been plausibly mediated by soft selective sweeps and/or polygenic adaptations rather

than by hard selective sweeps. For this purpose, we used a combination of composite-likelihood and gene network-based methods to detect adaptive loci in introgressed chromosomal segments from Tibetan WGS data and to shortlist those presenting Denisovan-like derived alleles that participate to the same functional pathways and are absent in populations of African ancestry, which are supposed to do not have experienced Denisovan admixture. According to this approach, we identified multiple genes putatively involved in archaic introgression events and that, especially as regards *TBC1D1*, *RASGRF2*, *PRKAG2*, and *KRAS*, have plausibly contributed to shape the adaptive modulation of angiogenesis and of certain cardiovascular traits in high-altitude Himalayan peoples. These findings provided unprecedented evidence about the complexity of the adaptive phenotype evolved by these human groups to cope with challenges imposed by hypobaric hypoxia, offering new insights into the tangled interplay of genetic determinants that mediates the physiological adjustments crucial for human adaptation to the high-altitude environment.

## Introduction

The scientific community currently agrees that the *Homo sapiens* species experienced admixture with extinct Hominins since traces of such inbreeding events are still detectable in the genomes of modern humans (*Gouy and Excoffier, 2020*). In fact, people belonging to non-African populations show 1–2% of Neanderthal ancestry (*Green et al., 2010*; *Prüfer et al., 2014*), while Melanesians and East-Asians present 3% and 0.2% of Denisovan ancestry, respectively (*Reich et al., 2010*; *Meyer et al., 2013*; *Prüfer et al., 2014*; *Racimo et al., 2017*). Despite evidence supporting selection against introgressed alleles has been collected (*Simonti et al., 2016*; *Racimo et al., 2017*; *McArthur et al., 2021*), some of the genomic segments showing signatures ascribable to archaic introgression were also proved to have been targeted by natural selection in modern human populations, thus providing examples for the occurrence of adaptive introgression (AI) events (*Racimo et al., 2017*).

So far, several studies have indeed identified introgressed archaic alleles at high frequency in human genes involved in metabolism or in the response to environmental conditions, such as temperature, sunlight, and altitude (*Prüfer et al., 2014*; *Vernot and Akey, 2014*; *Sankararaman et al., 2014*; *Huerta-Sánchez et al., 2014*; *Gittelman et al., 2016*; *Racimo et al., 2017*; *Enard and Petrov, 2018*; *Dannemann and Racimo, 2018*). Moreover, some genes that play a role in immune responses to pathogens are found to be characterized by a similar pattern of variability (*Laurent et al., 2011*; *Enard and Petrov, 2018*) and certain Neanderthal alleles have been shown to be associated with down-regulation of gene expression in brain and testes (*McCoy et al., 2017*; *Racimo et al., 2017*; *Dannemann and Racimo, 2018*). These works collectively attest how genetic variants introduced in the human gene pool by admixture with archaic species can significantly impact our biology by possibly comporting modifications in the modulation of several functional pathways. In particular, the high frequency of some archaic alleles in protein-coding and/or regulatory genomic regions suggests a possible adaptive role for Neanderthal and/or Denisovan variants, pointing to a further evolutionary mechanism having potentially contributed to the processes of human biological adaptation to different environmental and cultural settings. By introducing new alleles in the gene pool of a given population, admixture in fact provides a very rapid opportunity for natural selection to act on it (*Huerta-Sánchez et al., 2014*; *Jeong et al., 2014*; *Racimo et al., 2015*; *Hamid et al., 2021*) and this is supposed to have likely occurred during the evolutionary history of *H. sapiens*, particularly after the last Out of Africa migration in the late Pleistocene (*Sugden, 2018*; *Vahdati et al., 2022*). According to this view, gene flow from extinct Hominin species could have facilitated the adaptation of *H. sapiens* populations to peculiar Eurasian environments.

For instance, a Denisovan origin of the adaptive *EPAS1* haplotype, which confers reduced susceptibility to chronic mountain sickness to Tibetan and Sherpa highlanders (*Beall, 2007*; *Bigham et al., 2010*; *Yi et al., 2010*; *Peng et al., 2011*; *Xu et al., 2011*) is well established (*Huerta-Sánchez et al., 2014*; *Zhang et al., 2021*). However, the hard selective sweep experienced in high-altitude Himalayan populations by the *EPAS1* introgressed haplotype has been demonstrated to account only for an indirect aspect of their adaptive phenotype, which does not explain most of the physiological adjustments they evolved to cope with hypobaric hypoxia (*Gnecchi-Ruscone et al., 2018*). Therefore, how far gene flow between Denisovans and the ancestors of Tibetan/Sherpa peoples facilitated the evolution of other key adaptive traits of these populations remains to be elucidated.

To fill this gap, and to overcome the main limitation of most approaches currently used to test for AI (i.e., inferring archaic introgression and the action of natural selection separately by means of different algorithms, which increases the risk of obtaining biased results due to confounding variables), we assembled a dataset of whole-genome sequences (WGSs) from 27 individuals of Tibetan ancestry living at high altitude (*Cho et al., 2017*; *Jeong et al., 2018*) and we analysed it using a composite-likelihood method specifically developed to detect AI events at once (*Setter et al., 2020*). Notably, this method was designed to recognize AI mediated by subtle selective events (as those involved in polygenic adaptation) and/or soft selective sweeps, which represent the evolutionary mechanisms that are supposed to have played a more relevant role than hard selective sweeps during the adaptive history of human groups characterized by particularly small effective population size, such as Tibetans and Sherpa (*Gnecchi-Ruscone et al., 2018*). Coupled with validation of the identified putative adaptive introgressed loci through (1) the assessment of the composition of gene networks made up of functionally related DNA segments presenting archaic derived alleles that are absent in human groups which are supposed to do not have experienced Denisovan admixture, such as African ones, (2) the confirmation that natural selection targeted these genomic regions in populations of Tibetan ancestry, and (3) the quantification of genetic distance between modern and archaic haplotypes, such an approach provided new evidence about the biological functions that have mediated high-altitude adaptation in Himalayan populations and that have been favourably shaped by admixture of their ancestors with Denisovans.

## Results

### Spatial distribution of genomic variation and ancestry components of Tibetan samples

After quality control (QC) filtering of the available WGS data, we obtained a dataset made up of 27 individuals of Tibetan ancestry characterized for 6,921,628 single-nucleotide variants (SNVs). To assess whether this dataset represents a reliable proxy for the genomic variation observable in high-altitude Himalayan populations, we merged it with genome-wide genotyping data for 1086 individuals of East-Asian ancestry belonging to both low- and high-altitude groups (*Gnecchi-Ruscone et al., 2017*; *Landini et al., 2021*). We thus obtained an extended dataset including 231,947 SNVs (*Supplementary file 1a*), which was used to perform population structure analyses.

Results from ADMIXTURE and principal components analysis (PCA) were found to be concordant with those described in previous studies (*Jeong et al., 2014*; *Gnecchi-Ruscone et al., 2017*; *Gnecchi-Ruscone et al., 2018*; *Yang et al., 2021*). According to the ADMIXTURE model showing the best predictive accuracy ($K = 7$) (*Figure 1—figure supplement 1*), the examined WGS exhibited a predominant genetic component that was appreciably represented also in other populations speaking Tibeto-Burman languages, such as Tu, Yizu, Naxi, Lahu, and Sherpa (*Figure 1A* and *Figure 1—figure supplement 2*). Such a component reached an average proportion of around 78% in individuals of Tibetan ancestry from Nepal included in the extended dataset, as well as of more than 80% in the subjects under investigation, who live in the Nepalese regions of Mustang and Ghorka (*Figure 1A, B*). This suggests that after their relatively recent migration in Nepalese high-altitude valleys, these communities might have experienced a higher degree of isolation and genetic drift with respect to populations that are still settled on the Tibetan Plateau, in which the same ancestry fraction did not exceed 64% (*Figure 1A, B*). Nevertheless, the overall ADMIXTURE profile of the considered WGS appeared to be quite comparable to those inferred according to genome-wide genotyping data for other Tibetan populations (*Figure 1A, B*, *Figure 1—figure supplement 2*). Similarly, PCA pointed to the expected divergence of Tibetan and Sherpa high-altitude groups from the cline of genomic variation of East-Asian lowland populations (*Abdulla et al., 2009*; *Jeong et al., 2014*; *Gnecchi-Ruscone et al., 2017*; *Zhang et al., 2017*; *Wang et al., 2022*; *Figure 1C*). Remarkably, the WGS under investigation clustered within the bulk of genome-wide data generated for other groups from the Tibetan Plateau, thus supporting their representativeness as concerns the overall genetic background of high-altitude Himalayan populations.

### Detecting putative AI signatures in Tibetan genomes

To identify genomic regions showing signatures putatively ascribable to AI events, we scanned Tibetan WGS with the *VolcanoFinder* algorithm and we computed the composite likelihood ratio (LR)

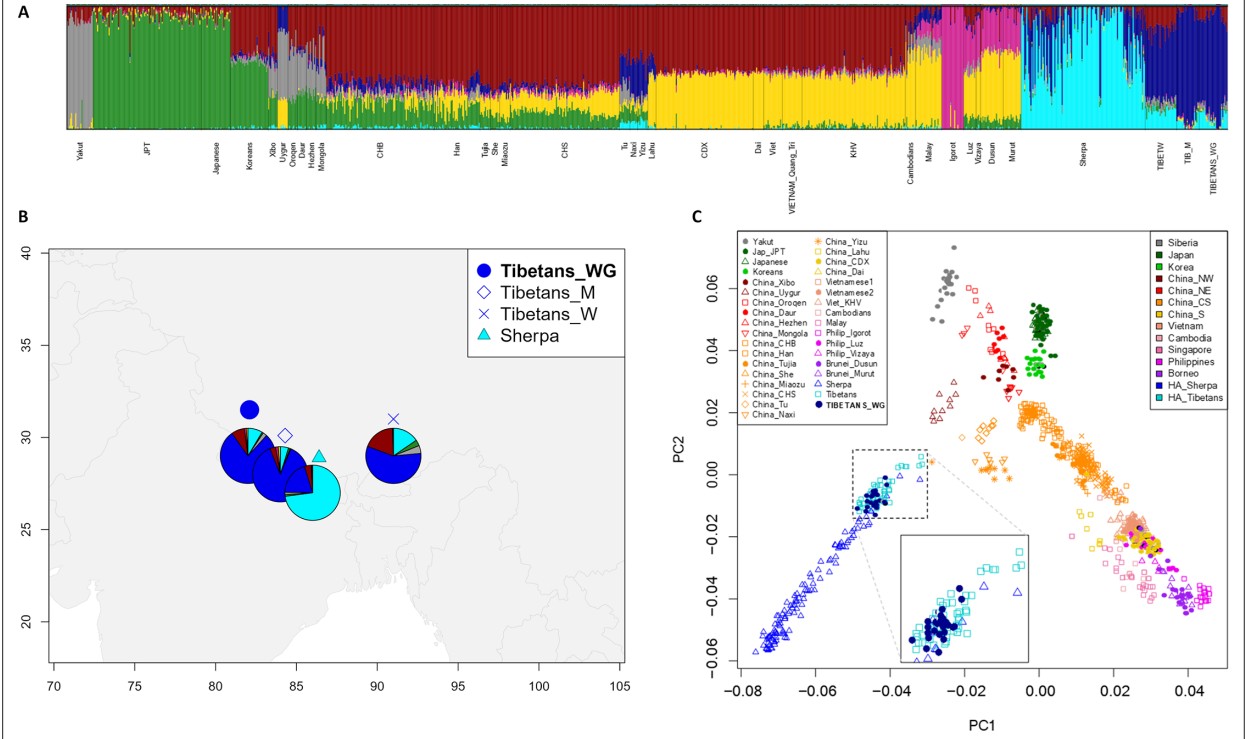

**Figure 1.** Population structure analyses performed on the extended dataset including Tibetan, Sherpa, and lowland East-Asian individuals. (**A**) Admixture analysis showed the best predictive accuracy when seven (*K* = 7) population clusters were tested. Populations included in the dataset are labelled according to population names and acronyms reported in *Supplementary file 1a*. (**B**) Map showing geographic location and admixture proportions at *K* = 7 of the high-altitude groups included in the extended dataset. The label *Tibetans_WG* indicates whole-genome sequence data for individuals of Tibetan ancestry analysed in the present study. Additional information about the considered samples (e.g., number of individuals per group, reference study, and used abbreviations) are reported in *Supplementary file 1a*. (**C**) Principal components analysis (PCA) plot considering PC1 vs PC2 and summarizing genomic divergence between high-altitude Tibetan/Sherpa people and the cline of variation observable for lowland East-Asian populations. The enlarged square displays clustering between Tibetan samples sequenced for the whole genome (i.e., blue dots) and Tibetan samples characterized by genome-wide data (i.e., light-blue squares).

The online version of this article includes the following figure supplement(s) for figure 1:

**Figure supplement 1.** Scatterplot showing the number of possible population clusters (*K*) tested by the different ADMIXTURE runs performed and the cross-validation (CV) errors associated to them.

**Figure supplement 2.** Admixture analyses performed on the extended dataset for *K* = 2 to *K* = 12.

and −log*α* statistics for each polymorphic site (*Setter et al., 2020*). We then considered the most significant results by focusing on loci showing LR values falling in the positive tail (i.e., top 5%) of the obtained distribution (see Materials and methods, *Supplementary file 1b*).

According to such an approach, we were first able to recapitulate the AI event previously described for the *EPAS1* gene (*Figure 2—figure supplement 1A*; *Huerta-Sánchez et al., 2014*; *Hu et al., 2017*; *Zhang et al., 2021*). In fact, this chromosomal interval was found to be characterized by a remarkable number of variants (*N* = 19) showing significant LR scores, as well as by high overall values of −log*α* (*Figure 2—figure supplement 1A*), suggesting, respectively, the plausible archaic origin of many alleles at this gene and an appreciable action of natural selection on it. Five of these significant SNVs have been already described as Denisovan-like derived alleles at outstanding frequency (i.e., ranging between 0.96 and 1) in Tibetans but not in other modern human populations (*Supplementary file 1c*; *Zhang et al., 2021*). On the contrary, at the genomic region encompassing *EGLN1* (which we have considered as a negative control for AI, see Materials and methods) we detected high −log*α* values coupled with a low number of SNVs (*N* = 3) showing significant LR scores, with only one being remarkably above the adopted significance threshold (*Figure 2—figure supplement 1B* and *Supplementary file 1d*). Overall, these findings are concordant with evidence from literature that suggest adaptive evolution of both *EPAS1* and *EGLN1* loci in high-altitude Himalayan populations (*Yang et al.,*

*2017*; *Liu et al., 2022*), although only the former was proved to have been impacted by archaic introgression (*Huerta-Sánchez et al., 2014*; *Hu et al., 2017*; *Zhang et al., 2021*).

Moreover, we were able to confirm other introgression signatures previously inferred from WGS data for populations of Tibetan ancestry, such as those involving the *PRKCE* gene and the *MIRLET7BHG* long non-coding region, which are located in the overlapping upstream chromosomal intervals, respectively, of *EPAS1* and *PPARA* (*Figure 2—figure supplement 2A, B*). In line with what reported for *EPAS1*, also *PPARA* has been already proposed to play a role in the modulation of high-altitude adaptation of Himalayan human groups (*Simonson et al., 2010*; *Horscroft et al., 2017*; *Zhang et al., 2021*). Interestingly, AI signatures identified by *VolcanoFinder* in the *MIRLET7BHG* locus extended also in the *PPARA* gene, as well in its downstream region (*Figure 2—figure supplement 2A*), supporting the findings described by *Hu et al., 2017*. Finally, we observed patterns comparable to those at *EPAS1* and *PPARA* for 10 additional genomic regions that were differentially pointed out by previous studies as Tibetan and/or Han Chinese DNA segments potentially carrying introgressed Denisovan alleles (*Hu et al., 2017*; *Browning et al., 2018*; *Zhang et al., 2021*; *Figure 2A, B*, *Figure 2—figure supplement 3A*, *Supplementary file 1e*).

## Validating genomic regions affected by archaic introgression

To validate signatures of archaic introgression at the candidate AI loci identified with *VolcanoFinder*, we relied on the approach described by *Gouy and Excoffier, 2020*. In detail, we used the *Signet* algorithm to identify networks of genes participating to the same functional pathway and presenting archaic-like (i.e., Denisovan) derived alleles observable in the putative admixed group (i.e., Tibetans) but not in an outgroup of African ancestry (i.e., Yoruba, YRI), by assuming that only Eurasian *H. sapiens* populations experienced Denisovan admixture (see Materials and methods).

After having crosschecked results from the *VolcanoFinder* and *Signet* analyses, we identified six gene networks that turned out to be consistently significant across all the *Signet* runs performed and that included a total of 15 genes pointed by *VolcanoFinder* as candidate AI loci (*Supplementary file 1f*). Four of these loci composed the gene network overall ascribable to the *Pathways in Cancer* biological functions, which included also the *EPAS1* positive control for AI (*Figure 3A*). Most of the other genes supported by both the analyses were instead observed in significant networks belonging to the *Ras signalling* and *AMPK signalling* pathways (*Supplementary file 1f*).

Interestingly, gene networks belonging to the *Pathways in Cancer* and *Ras signalling pathway* appeared to be tightly related from a functional perspective because included oncogenes that promote the initiation and progression of tumour growth by stimulating cell proliferation and angiogenesis (*Kranenburg et al., 2004*). In particular, according to the Kyoto Encyclopaedia of Genes and Genomes (KEGG) database *KRAS* and *RASGRF2* genes from the *Ras signalling* network were found to contribute also to the *Pathways in Cancer* functions (*Figure 3A*), especially by interacting with the identified candidate introgressed loci *PLCB1*, *RASGRP2*, *DAPK1*, *MAPK1*, *FOS*, and *VEGFA* to modulate the *VEGF signalling pathway*, which is activated in hypoxic conditions and induces the transcription of genes that promote angiogenesis (*Figure 3A*; *Maxwell and Ratcliffe, 2002*; *Kranenburg et al., 2004*).

Also, the *AMPK signalling pathway* is known to be activated in different cell types by stresses such as deprivation of oxygen and/or glucose, leading to the inhibition of energy-consuming biosynthetic pathways (e.g., protein and glycogen synthesis) and to the activation of ATP-producing catabolic pathways, such as fatty acid oxidation and glycolysis (*Kanehisa and Goto, 2000*; *Chen et al., 2018*; *Dengler, 2020*).

No significant gene networks involving the *EGLN1* genomic region considered as a negative control for AI were instead reconstructed with the *Signet* approach (*Supplementary file 1f*), suggesting that the very few variants at this locus that showed *VolcanoFinder* LR scores above the adopted significant threshold might represent false positive results (*Figure 2—figure supplement 1B* and *Supplementary file 1d*).

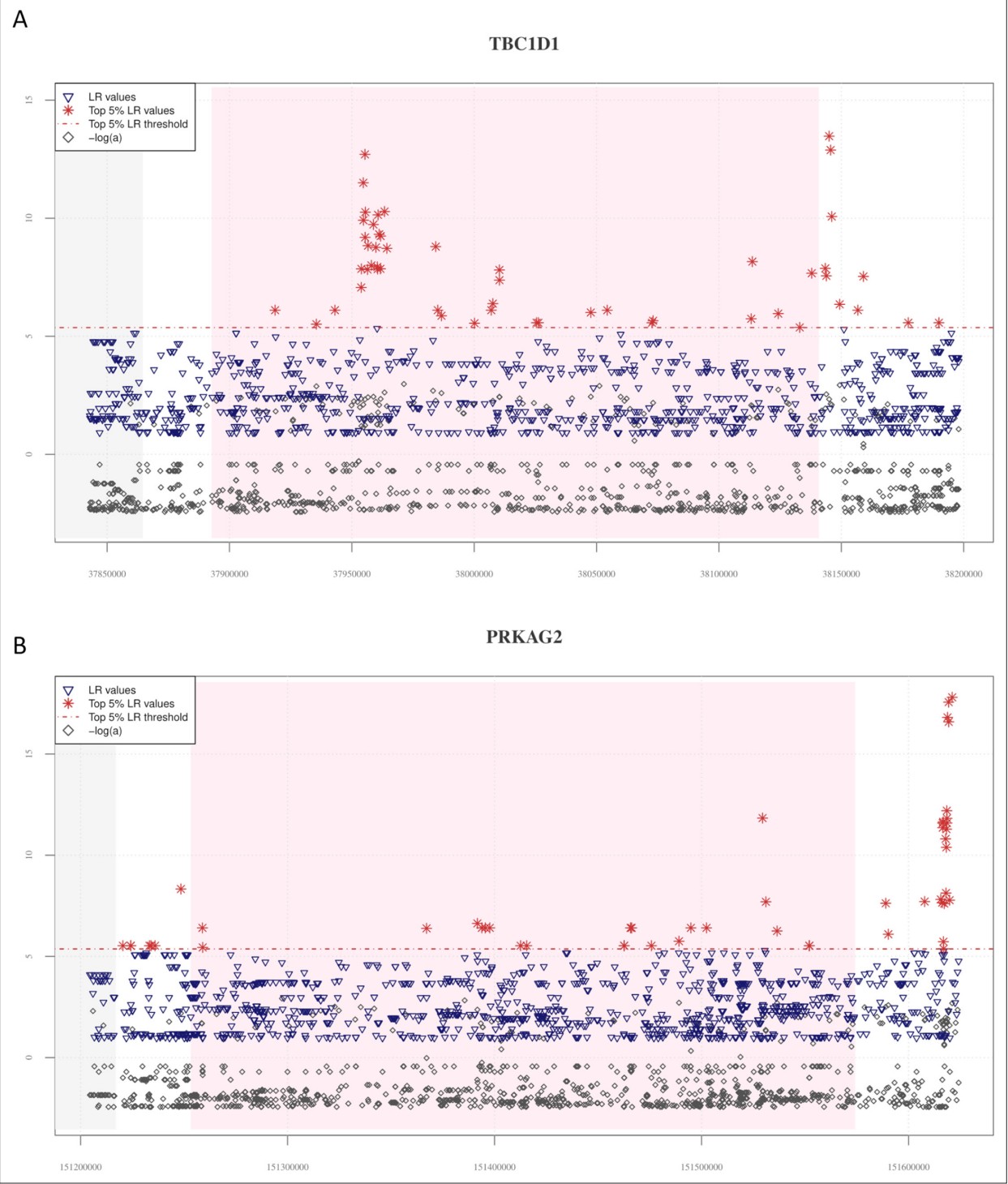

**Figure 2.** Distribution of *VolcanoFinder* statistics suggestive of putative adaptive introgrossed loci across the *TBC1D1* and *PRKAG2* genomic regions. On the *x*-axis are reported genomic positions of each single-nucleotide variant (SNV), while on the *y*-axis are displayed the related statistics obtained. Pink background indicates the chromosomal interval occupied by the considered genes, while the grey background identifies those genes (i.e., *PGM2* in the *TBC1D1* downstream genomic region and the *RHEB* gene in the upstream *PRKAG2* region) possibly involved in regulatory transcription mechanisms. The dashed red line identifies the threshold set to filter for significant likelihood ratio (LR) values (i.e., top 5% of LR values). For both these genomic regions, the distribution of LR and $-\log\alpha$ are concordant with those observed at the *EPAS1* positive control for adaptive introgression (AI). (**A**) A total of 50 significant LR values (red stars) and $-\log\alpha$ (grey diamonds) values resulted collectively elevated in both the *TBC1D1* gene and its downstream genomic regions. A remarkable concentration of significant LR values characterizing 19 SNVs was especially observable in the first portion

*Figure 2 continued on next page*

Figure 2 continued

of the gene. (**B**) The entire *PRKAG2* genomic region was found to comprise 46 SNVs showing significant LR values, with the greatest peaks being located in the downstream region associated to such gene. Peaks detected for the LR statistic are accompanied by peaks of −logα values.

The online version of this article includes the following figure supplement(s) for figure 2:

**Figure supplement 1.** Distribution of *VolcanoFinder* statistics across the *EPAS1* and *EGLN1* positive and negative controls for adaptive introgression.

**Figure supplement 2.** Distribution of *VolcanoFinder* statistics across *MIRLET7BHG, PPARA*, and *PRKCE* genes.

**Figure supplement 3.** Distribution of *VolcanoFinder* statistics across the *RASGRF2* candidate adaptive introgression (AI) gene.

**Figure supplement 4.** Distribution of *VolcanoFinder* statistics across the *KRAS* candidate adaptive introgression (AI) gene.

## Shortlisting introgressed genomic regions characterized by adaptive evolution

To further shortlist the most robust candidate genes involved in AI events, we applied the *LASSI* algorithm to phased Tibetan WGS data with the aim of searching for genomic signatures ascribable to the action of natural selection (*Harris and DeGiorgio, 2020*) (see Materials and methods).

This enabled us to confirm the strong selective events occurred at the *EPAS1* and *EGLN1* genes, as previously reported by multiple studies conducted on high-altitude Himalayan populations (*Beall et al., 2010*; *Yi et al., 2010*; *Simonson et al., 2010*; *Horscroft et al., 2017*; *Zhang et al., 2021*), as well as to corroborate adaptive evolution of some of the genes pointed out by both *VolcanoFinder* and *Signet* analyses. In fact, several chromosomal intervals associated to these loci presented values of the computed *T* statistic that fall within the top 5% of the related distribution (*Figure 4C, D*, *Figure 4—figure supplements 1 and 2C, D*, *Figure 4—figure supplement 3C, D*).

More in detail, in addition to *EPAS1*, genomic windows associated to the *DAPK1, GNG7, AK5, TBC1D1, PLCB1, RASGRF2*, and *PRKAG2* introgressed loci supported by both *VolcanoFinder* and *Signet* approaches were found to present scores within the top 5% of the *T* distribution, suggesting that their haplotype diversity was appreciably shaped by positive selection. Interestingly, adaptive evolution of the *TBC1D1* and *RASGRF2* genes has been previously proposed by studies conducted on different populations of Tibetan ancestry (*Peng et al., 2011*; *Zheng et al., 2023*).

## Estimating genetic distance between modern and archaic sequences

As a final step for prioritizing the most convincing AI genes supported by *VolcanoFinder*, *Signet*, and *LASSI* approaches, as well as to explicitly test whether the Denisovan human species represented a plausible source of archaic alleles for them, we merged Tibetan WGS data with those from low-altitude Han Chinese (CHB) and YRI populations sequenced by the 1000 Genomes Project (*Auton et al., 2015*), and with the Denisovan genome. We then used the *Haplostrip* algorithm (*Marnetto and Huerta-Sánchez, 2017*) to estimate genetic distance between modern and archaic haplotypes at the candidate AI genes reported in the previous paragraph. We especially considered genomic windows that included Denisovan-like derived alleles and that presented values of the likelihood *T* statistic supporting an adaptive evolution (see Materials and methods).

Among the tested putative AI loci, *TBC1D1, RASGRF2, PRKAG2*, and *KRAS* were found to present substantial proportions of Tibetan haplotypes that cluster close to the Denisovan sequence, thus showing the lowest numbers of pairwise differences with respect to it as compared with CHB or YRI haplotypes (*Figure 4A, B*, *Figure 4—figure supplement 2A, B*). More in detail, 61% of the *TBC1D1* Tibetan haplotypes turned out to be the nearest ones to the archaic sequence by entailing only two pairwise differences with respect to it (*Figure 4A*), while 29% of Tibetan haplotypes inferred for the *RASGRF2* gene were even identical to the Denisovan DNA (*Figure 4B*). Interestingly, both these chromosomal intervals were classified by the *LASSI* method as regions whose variation pattern was conformed with the soft selective sweep model, presenting three potential adaptive haplotypes (i.e., those haplotypes that plausibly carry putative advantageous alleles and thus increased in frequency due to positive selection). At each gene, one of these haplotypes was found to contain the Denisovan-like derived alleles that are completely absent in YRI (*Figure 4A, D*). A similar pattern was observed also for the considered *PRKAG2* and *KRAS* genomic windows (*Figure 4—figure supplement 2A, B*).

Overall, the distribution of similarities between modern and archaic haplotypes described for the four identified AI candidate loci appears to be comparable to that obtained for *EPAS1*, with the sole relevant distinction being represented by an even more pronounced differentiation between Tibetan

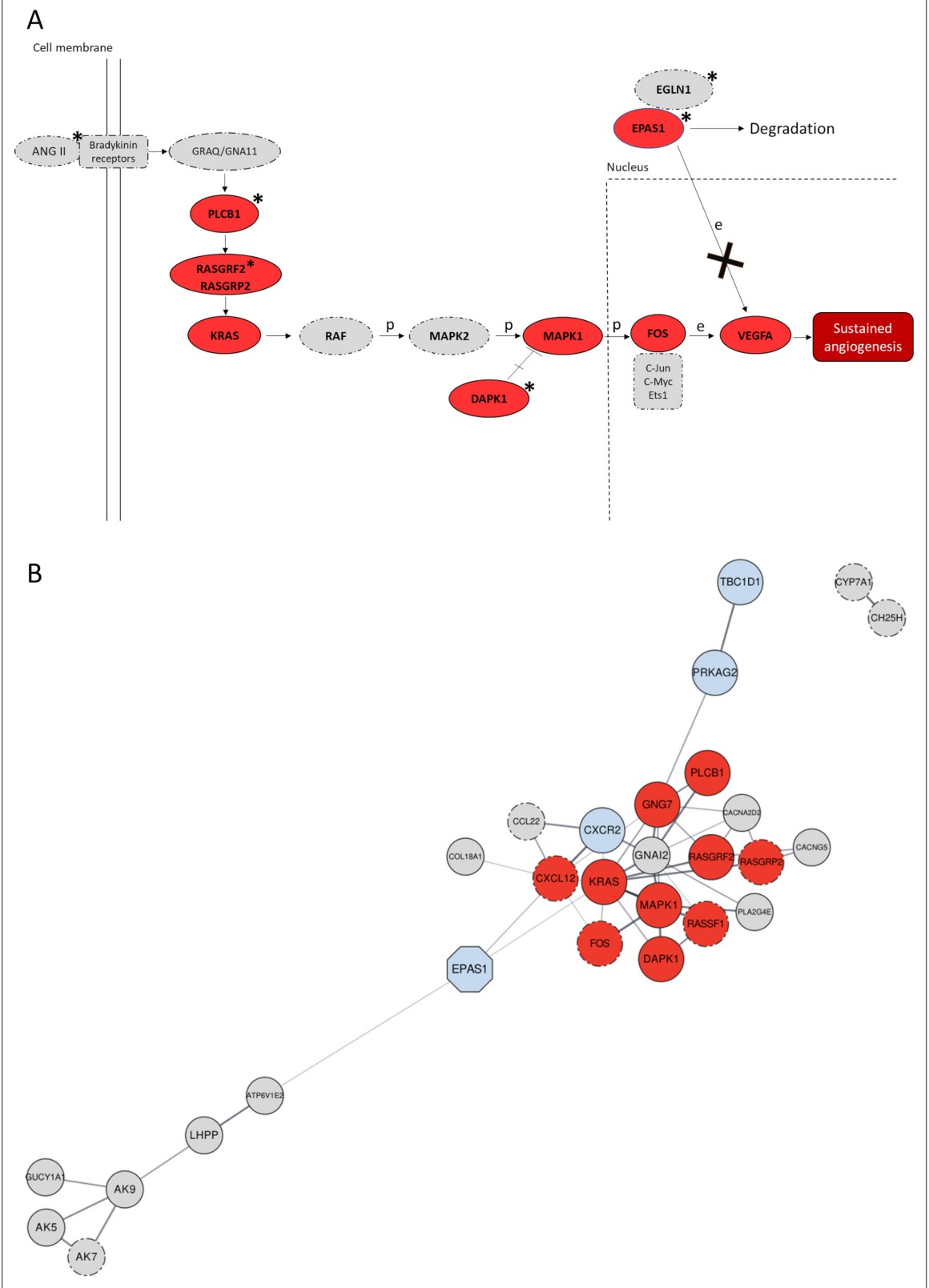

**Figure 3.** Significant gene networks including Denisovan-like derived alleles according to the *Signet* analysis. (**A**) Schematic representation of the activation of the *RAS/MAPK*(*ERK*) axis after interaction of the bradykinin receptors with their ligands (e.g., ANG II) within the framework of the *Pathways in Cancer* network. Genes supported by both *Signet* (i.e., belonging to the significant network associated to *Pathways in cancer*) and *VolcanoFinder* (i.e., including at least a single-nucleotide variant (SNV) showing likelihood ratio (LR) value within top 5% of the obtained results) analyses as potentially

*Figure 3 continued on next page*

Figure 3 continued

introgressed loci, are highlighted in red and present solid outline. Grey circles with dotted-dashed contour instead indicate genes supported only by *Signet*, while loci marked with stars are those including genomic windows showing *LASSI T* statistic within top 5% of the related distribution. After the interaction between ANG II (active enzyme angiotensin II) and bradykinin receptors, activation of the Ras protein encoded by *KRAS* mediated by RAS-GTPases (e.g., *RASGRF2*) comports a series of phosphorylation reactions that eventually promotes angiogenesis (*Kranenburg et al., 2004*). In detail, phosphorylation of the MAPK1 protein and prevention of MAPK1-DAPK-1-dependent apoptosis leads to increased MAPK1 activity (*Kanehisa and Goto, 2000*; *Stevens et al., 2007*) that causes improved *FOS* mRNA expression (*Monje et al., 2005*). *FOS* together with other proteins (e.g., Jun) forms the AP-1 transcription factor, which bounds to the *VEGF* promoter region upregulating its expression in endothelial cells (*Catar et al., 2013*) and sustaining angiogenesis when the hypoxia inducible factor 1 (HIF-1) signalling cascade is inhibited (*Lorenzo et al., 2014*). (B) Gene network built by setting co-expression as force function and by displaying the entire set of genes identified by the *Signet* algorithm as belonging to significant pathways including Denisovan-like derived alleles. Genes whose variation pattern was supported by both *VolcanoFinder* and *Signet* analyses (e.g., *TBC1D1*) as shaped by archaic introgression are displayed with a solid black outline. The *EPAS1* positive control locus that has been previously proved to have mediated adaptive introgression in Tibetan populations was represented as light-blue octagonal. Genes included in pathways involved in angiogenesis (e.g., *RASGRF2*) and/or activated in hypoxic conditions (e.g., *PRKAG2*) are reported as dark red and light-blue circles, respectively, while the remining fraction of significant genes are represented as light-grey circles. The closeness or the distance between all nodes reflects the tendency to be co-expressed with each other and all the connections inferred are characterized by a confident score ≥0.7.

and CHB patterns plausibly ascribable to the occurrence of a hard (rather than soft) selective sweep at *EPAS1* (*Figure 4—figure supplement 3A*), as previously proposed (*Simonson et al., 2010*; *Huerta-Sánchez et al., 2014*). In fact, at *EPAS1* the haplotype carrying the lowest number of pairwise differences ($N = 3$) with respect to the archaic one belongs to the Tibetan population, in which it reached 74% of frequency, being instead absent in all the others modern groups considered (*Figure 4—figure supplement 3A*). Conversely, the *EGLN1* genomic window showing the highest *T* score according to *LASSI* analysis was characterized by an opposite pattern, with CHB haplotypes being overrepresented among those with the lowest number of pairwise differences ($N = 3$) with respect to the Denisovan genome (*Figure 4—figure supplement 3B*). Moreover, the sole *EGLN1* putative adaptive haplotype inferred by *LASSI* for Tibetans was among those presenting the highest number of differences as compared with the archaic sequence, being characterized exclusively by alleles that are not observed in the Denisovan genome and thus suggesting that natural selection targeted modern rather than archaic *EGLN1* variation (*Figure 4—figure supplement 3B*).

## Discussion

In the present study, we aimed at investigating how far gene flow between the Denisovan archaic human species and the ancestors of modern populations settled in high-altitude regions of the Himalayas contributed to the evolution of key adaptive traits of these human groups, in addition to having conferred them reduced susceptibility to chronic mountain sickness (*Huerta-Sánchez et al., 2014*). For this purpose, we used WGS data from individuals of Tibetan ancestry to search for genomic signatures ascribable to AI mediated by weak selective events rather than by hard selective sweeps, under the assumption that soft sweeps and/or processes of polygenic adaptation are more likely to have occurred in such remarkably isolated and small effective population size groups (*Gnecchi-Ruscone et al., 2018*).

By assembling a large genome-wide dataset including both low- and high-altitude populations, we first framed the available WGS data into the landscape of East-Asian genomic variation. This confirmed that the genomes under investigation are well representative of the overall profiles of ancestry components observable in high-altitude Himalayan peoples (*Figure 1A, B*). In fact, the considered individuals were found to show close genetic similarity to other populations of Tibetan ancestry (*Jeong et al., 2014*) and to Sherpa people from Nepal (*Gnecchi-Ruscone et al., 2017*), as well as to appreciably diverge from the cline of variation of lowland East-Asians (*Abdulla et al., 2009*; *Figure 1C*).

Based on this evidence, we submitted Tibetan WGS to a pipeline of analyses that implemented multiple independent approaches aimed at identifying genomic regions characterized by signatures putatively ascribable to AI events and by tight functional correlations with each other. According to such a rationale, we shortlisted the candidate introgressed loci that most likely contribute to the same adaptive trait by searching for chromosomal intervals including loci simultaneously showing: (1) significant LR scores computed by the *VolcanoFinder* algorithm (*Figure 2A, B*, *Figure 2—figure*

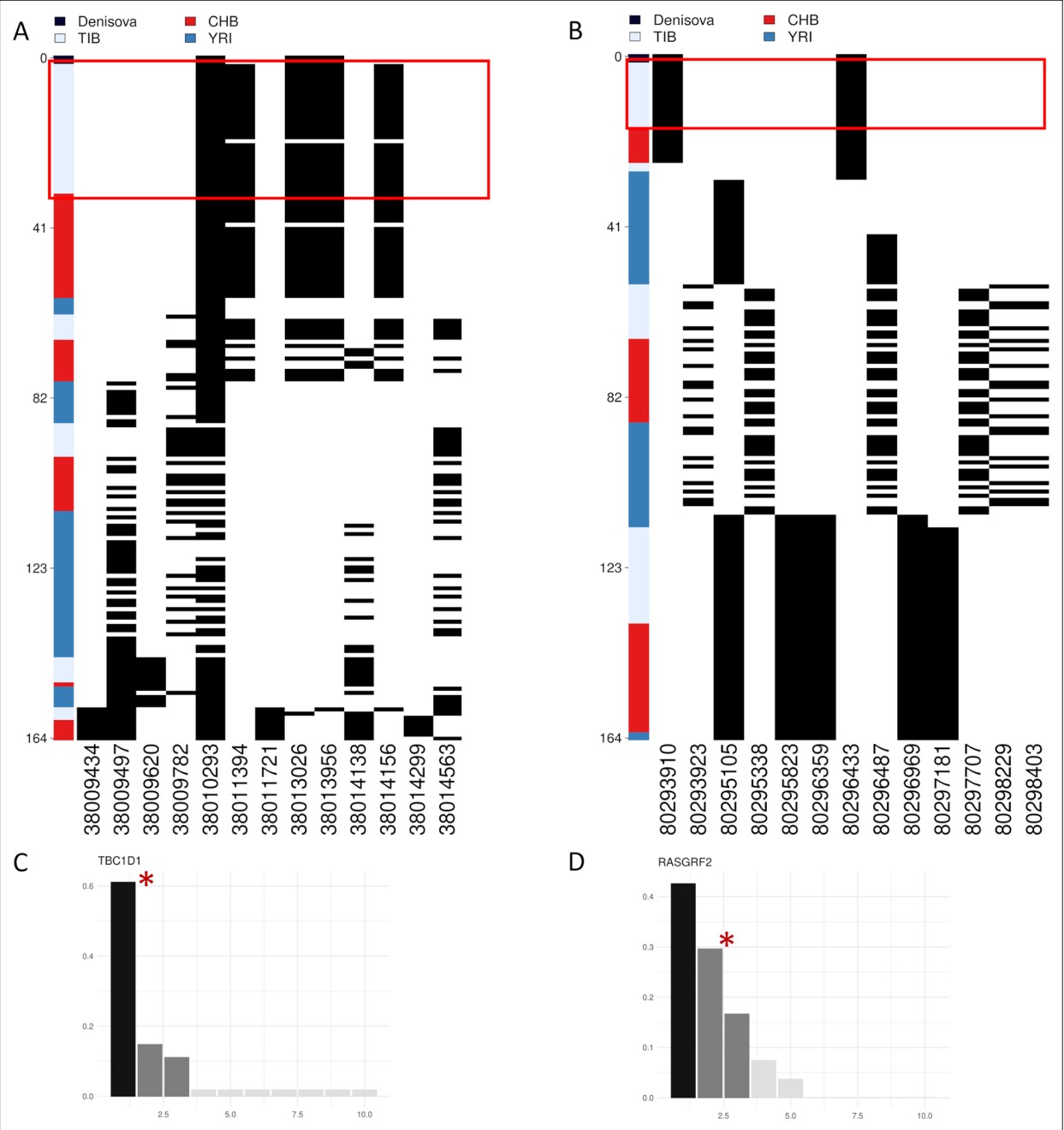

**Figure 4.** Representation of genetic distances between modern and archaic haplotypes and barplots showing haplotype frequency spectra for *TBC1D1* and *RASGRF2* candidate adaptive introgression (AI) genes. Haplotypes are reported in rows, while derived (i.e., black square) and ancestral (i.e., white square) alleles are displayed in columns. Haplotypes are ranked from top to bottom according to their number of pairwise differences with respect to the Denisovan sequence. (**A**) Heatmap displaying divergence between Tibetan, CHB and YRI *TBC1D1* haplotypes with respect to the Denisovan genome. A total of 33 *TBC1D1* haplotypes (i.e., 61% of the overall haplotypes inferred for such a region) belonging to individuals with Tibetan ancestry are plotted in the upper part of the heatmap thus presenting the smallest number of pairwise differences with respect to the Denisovan sequence. (**B**) Heatmap displaying divergence between Tibetan, CHB and YRI *RASGRF2* haplotypes with respect to the Denisovan genome. A total of 16 Tibetan haplotypes in the *RASGRF2* genomic region present no differences with respect to the Denisovan sequence. As regards barplots, on the x-axis are reported the haplotypes detected in the considered genomic windows, while on the y-axis is indicated the frequency for each haplotype. The black and dark-grey bars indicate the more frequent haplotypes (i.e., the putative adaptive haplotypes inferred by the *LASSI* method), while red stars mark those haplotypes carrying Denisovan-like derived alleles. (**C**) *TBC1D1* haplotype frequency spectrum. The *TBC1D1* gene presents a haplotype pattern qualitatively comparable to that observed at *EPAS1* (*Figure 4—figure supplement 3A*), with a predominant haplotype carrying archaic derived alleles and reaching elevated frequencies in Tibetan populations. In line with this observation, such a pattern was inferred by *LASSI* as conformed with a non-neutral evolutionary scenario, even if it seems to be characterized by a soft rather than a hard selective sweep due to the occurrence of three sweeping

*Figure 4 continued on next page*

Figure 4 continued

haplotypes. (**D**) *RASGRF2* haplotype frequency spectrum. A soft selective sweep was inferred also for the considered *RASGRF2* genomic window, although frequencies reached by the sweeping haplotypes turned out to be more similar with each other. The second most represented haplotype was that carrying the archaic derived alleles and, reached a frequency of 29% in the Tibetan group.

The online version of this article includes the following figure supplement(s) for figure 4:

**Figure supplement 1.** Haplotype frequency spectra of the top windows detected as adaptively evolved by *LASSI* in the *EPAS1* and *EGLN1* genomic regions.

**Figure supplement 2.** Representation of genetic distances between modern and archaic haplotypes.

**Figure supplement 3.** Representation of genetic distances between modern and archaic haplotypes.

*supplements 3 and 4*), (2) Denisovan-like derived alleles belonging to significant networks of functionally related genes reconstructed with the *Signet* method and completely absent in populations of African ancestry (*Figure 3A, B*, *Supplementary file 1f*), (3) signatures ascribable to the action of natural selection as pointed out by *LASSI* analysis (*Figure 4C, D*, *Figure 4—figure supplement 2C, D*), and (4) haplotypes more similar to the Denisovan ones rather than to those observed in other modern human populations, as depicted by the *Haplostrips* approach (*Figure 4A, B*; *Figure 4—figure supplement 2A, B*).

Overall, in addition to *EPAS1*, which we considered as a positive control for AI, a total of 18 genes encompassed within the putative AI chromosomal intervals identified by the *VolcanoFinder* method (*Supplementary file 1e*) were found to have been previously proposed as genomic regions impacted by introgression of Denisovan alleles in Tibetan and/or Han Chinese populations (*Huerta-Sánchez et al., 2014*; *Hu et al., 2017*; *Browning et al., 2018*; *Yang et al., 2017*; *Zhang et al., 2021*). Among them, *PPARA*, *PRKCE*, and *TBC1D1* (*Figure 2A*, *Figure 2—figure supplement 2A, B*) were also specifically suggested to have played an adaptive role in high-altitude groups from the Tibetan Plateau (*Simonson et al., 2010*; *Peng et al., 2011*; *Horscroft et al., 2017*; *Arciero et al., 2018*; *Deng et al., 2019*; *Zhang et al., 2021*; *Zheng et al., 2023*). Interestingly, *PPARA* encodes for a nuclear transcription factor whose decreased expression in the myocardium of rats exposed to hypoxia seems to contribute to the maintenance of the correct heart contractile function despite such a stressful condition (*Cole et al., 2016*). Similarly, the PRKCE protein kinase C has been demonstrated to exert a cardio-protective role against ischemic injury (*Scruggs et al., 2016*). Moreover, *TBC1D1* encodes for a protein whose serine phosphorylation sites are targeted by AMP-activated protein kinases (AMPK) after the activation of the *AMPK signalling pathway* as a result of the increase cellular AMP/ATP ratio caused by stresses that induce a lower ATP production (e.g., deprivation of oxygen and/or glucose) or that accelerate ATP consumption (e.g., intense muscle contraction) (*Kanehisa and Goto, 2000*; *Vichaiwong et al., 2010*). In addition, another member of the *AMPK signalling pathway*, *PRKAG2*, was suggested by both *VolcanoFinder* analysis and literature data to present putative introgressed Denisovan alleles in Tibetan populations (*Figure 2B*; *Zhang et al., 2021*). Mutations at this locus are known to cause the *PRKAG2 cardiac syndrome*, an inherited disease characterized by ventricular pre-excitation, supraventricular arrhythmias, and cardiac hypertrophy (*Zhang et al., 2013*; *Porto et al., 2016*). Dysregulation of AMPK activity mediated by reduction in *PRKAG2* expression and leading to the impairment of glycogen metabolism in cell cultures has been proposed as a possible cause for the development of this pathological condition (*Zhang et al., 2013*). Conversely, enhanced activation of the *AMPK signalling pathway* during pregnancy coupled with *PRKAG2* overexpression was observed in the placenta of women living at high altitudes when compared with women living in low-altitude regions (*Lorca et al., 2021*). Finally, high-altitude individuals of Tibetan ancestry were found to exhibit reduced incidence of major adverse cardiovascular events with respect to low-altitude controls possibly indicating the involvement of protective cardiac mechanisms in the modulation of high-altitude adaptations as previously proposed (*Kolár and Ostádal, 2004*; *Mallet et al., 2018*; *Lei et al., 2024*). We can thus speculate that adaptive evolution at the *PPARA*, *PRKCE*, *TBC1D1*, and *PRKAG2* genomic regions in Tibetans might have contributed to the development of protective mechanisms that reduce cardiovascular risk associated to the hypoxic stress.

In addition to these genes, other two of the identified candidate AI loci (i.e., *RASGRF2* and *KRAS*) have been previously proved to have been targeted by natural selection in Tibetan populations (*Peng et al., 2011*) or to present putative introgressed archaic alleles (*Hu et al., 2017*; *Browning et al.,*

*2018*; *Figure 2—figure supplements 3 and 4*). The proteins encoded by these loci are strictly related from a functional perspective, with the Ras protein specific guanine nucleotide releasing factor 2 representing a calcium-regulated nucleotide exchange factor that activates the RAS protein codified by the proto-oncogene *KRAS* (*Kanehisa and Goto, 2000*; *Sayers et al., 2022*).

Despite evidence reported in literature seem to corroborate *VolcanoFinder* results that indicate *PPARA* and *PRKCE* as putatively implicated in AI events experienced by Tibetan ancestors, only *EPAS1*, *TBC1D1*, *RASGRF2*, *PRKAG2*, and *KRAS* loci were finally retained after the adopted filtering procedure to represent the most reliable candidate AI loci.

Archaic introgression at these genomic regions was first confirmed by *Signet* analysis, with *TBC1D1* and *PRKAG2* being included in a significant gene network belonging to the *AMPK signalling pathway*, while *RASGRF2* and *KRAS* participate to that related to the *Ras signalling pathway* (*Supplementary file 1f*). The same analysis pointed to *EPAS1* as a member of a significant network belonging to the *Pathways in cancer*, a complex group of biological functions such as those involved in Ras, MAPK, VEGF, and HIF-1 signalling cascades (*Kanehisa and Goto, 2000*; *Figure 3A* and *Supplementary file 1f*). These findings emphasize a link between biological mechanisms activated within the context of hypoxic tumour microenvironments in different types of cancers and those involved in high-altitude adaptations, especially as concerns functions that might underlie the improved angiogenesis observed in Tibetan and Sherpa individuals (*Gnecchi-Ruscone et al., 2018*; *Calderón - Gerstein and Torres - Samaniego, 2021*). For instance, accumulation of the HIF-1α transcriptional factor in the nucleus of cells close to hypoxic tumour masses comports the activation of diverse biological responses such as the formation of dense capillary structures that permit oxygen and nutrients supplies to cancer cells, thus determining tumour progression and/or treatment failure (*Brahimi-Horn et al., 2007*; *Calderón - Gerstein and Torres - Samaniego, 2021*). In line with this evidence, the *Ras* and *MAPK/ERK signalling pathways* have been proposed to play a significant role in promoting angiogenesis by triggering *VEGF* expression, being possibly implicated in the adaptive response to hypoxia evolved by high-altitude populations (*Figure 3A*; *Kanehisa and Goto, 2000*; *Kranenburg et al., 2004*). Moreover, in the study by *Lorenzo et al., 2014* a gain-of-function mutation in the *EGLN1* gene was observed in Tibetans and was proved to enhance the catalytic activity of the HIF prolyl 4-hydroxylase 2 (*PHD2*) under hypoxic conditions. This alters the binding of HIF-2α (the isoform 2 of the inducible hypoxia transcriptional factor encoded by *EPAS1*) and negatively regulates the activation of the *HIF-1 signalling pathway* during hypoxia, eventually offering protection against the detrimental effects of prolonged polycythaemia. When HIF-2α exerts its transcriptional activities along with p300 protein and HIF-1β, it enhances *VEGF* expression and permits the activation of the *VEGF signalling pathway* (*Kanehisa and Goto, 2000*; *Rashid et al., 2021*). Accordingly, down-regulation of the *HIF-1 signalling pathway* comports the reduction of VEGF mRNA expression (*Greenberger et al., 2008*; *Zhang et al., 2018*). Coupled with these observations, results from the *Signet* analysis suggest that in individuals of Tibetan ancestry, when the *HIF-1 signalling pathway* is likely down-regulated in chronic hypoxic conditions (*Lorenzo et al., 2014*), adaptive changes at the *Ras/MAPK signalling pathways* could represent alternative biological mechanisms that in its place enable to sustain improved angiogenesis and thus permit adequate tissue oxygenation (*Figure 3A*).

The same five candidate genes, in addition to the *EGLN1* locus considered as a negative control for AI, were also confirmed by the *LASSI* method to have adaptively evolved in the studied populations (*Figure 4C, D*, *Figure 4—figure supplement 1A, B*, *Figure 4—figure supplements 2 and 3C, D*). In fact, several genomic windows in their associated chromosomal intervals and/or in their flanking regions presented positive values of the computed likelihood $T$ statistic, many of which falling in the top 5% of the related distribution, which indicate their non-neutral evolution. Interestingly, *TBC1D1*, *RASGRF2*, *PRKAG2*, *KRAS*, and *EPAS1* significant genomic windows pointed out by *LASSI* were also found to include Denisovan-like derived alleles that are completely absent in the YRI African population, suggesting that positive selection acted in Tibetans on haplotypes containing archaic introgressed variation. Such a scenario was further supported by the *Haplostrips* analysis, which revealed for all the loci mentioned above patterns of similarity between Tibetan and Denisovan haplotypes that are comparable to that observed for *EPAS1* (*Figure 4A, B*, *Figure 4—figure supplement 2A, B*), with the sole exception being represented by *EGLN1* (*Figure 4—figure supplement 1B*, *Figure 4—figure supplement 3B, D*, *Figure 4—figure supplement 3A*).

Overall, we collected multiple evidence supporting both the archaic origin and the adaptive role of variation at *TBC1D1*, *RASGRF2*, *PRKAG2*, and *KRAS* genes in populations of Tibetan ancestry. Genetic signatures at such loci are especially consistent with the hypothesis of adaptive events mediated by soft selective sweeps and/or polygenic mechanisms that involved haplotypes including both modern and archaic introgressed alleles. Therefore, the results obtained have succeeded in expanding the knowledge about AI events having involved the ancestors of modern high-altitude Himalayan populations and Denisovans and emphasized once more the complexity of the adaptive phenotype evolved by these human groups to cope with challenges imposed by hypobaric hypoxia. Accordingly, they offer new insights for future studies aimed at elucidating the molecular mechanisms by which several genes along with *TBC1D1*, *PRKAG2*, *RASGRF2*, and *KRAS* interact with each other and contribute to mediate physiological adjustments that are crucial for human adaptation to the high-altitude environment.

## Materials and methods

### Ethics

The University of Bologna Ethics Committee released approval (Prot. 205142, 12/9/2019) for the present study within the framework of the project 'Genetic adaptation and acclimatization to high altitude as experimental models to investigate the biological mechanisms that regulate physiological responses to hypoxia'. However, no new sampling campaign was performed in the context of the present study and all the genomic data analysed were publicly available. The informed consent for the 27 Tibetan WGS analysed here was previously obtained and declared in the *Ethics statement* section of the study by *Jeong et al., 2018*.

### Dataset composition and curation

The dataset used in the present study included WGS data for 27 individuals of Tibetan ancestry recruited from the high-altitude Nepalese districts of Mustang and Ghorka (*Jeong et al., 2018*). Although these subjects reside in Nepal, they have been previously proved to speak Tibetan dialects and to live in communities showing religious and social organizations proper of populations settled on the Tibetan Plateau, being also biologically representative of high-altitude Tibetan people (*Cho et al., 2017*). To filter for high-quality genotypes, the dataset was subjected to QC procedures using the software PLINK v1.9 (*Purcell et al., 2007*). In detail, we retained autosomal SNVs characterized by no significant deviations from the Hardy–Weinberg equilibrium ($p > 5.3 \times 10^{-9}$ after Bonferroni correction for multiple testing), as well as loci/samples showing less than 5% of missing data. Moreover, we removed SNVs with ambiguous A/T or G/C alleles and multiallelic variants, obtaining a dataset composed by 27 individuals and 6,921,628 SNVs. WGS data were finally phased with SHAPEIT2 v2.r904 (*Delaneau et al., 2013*) by applying default parameters and using the 1000 Genomes Project dataset as a reference panel (*Auton et al., 2015*) and HapMap phase 3 recombination maps.

### Population structure analyses

To assess representativeness, genetic homogeneity, and ancestry composition of Tibetan WGS included in the dataset, we performed genotype-based population structure analyses. For this purpose, we merged the unphased WGS dataset with genome-wide genotyping data for 34 East-Asian populations (*Gnecchi-Ruscone et al., 2017*; *Landini et al., 2021*) and we applied the same QC described above. The obtained extended dataset included 231,947 SNVs and was used to assess the degree of recent shared ancestry (i.e., identity-by-descent, IBD) for each pair of individuals. Identity-by-state (IBS) estimates were thus used to calculate an IBD kinship coefficient and a threshold of PI_HAT = 0.270 was considered to remove closely related subjects to the second degree (*Ojeda-Granados et al., 2022*). To discard variants in linkage disequilibrium (LD) with each other we then removed one SNV for each pair showing $r^2 > 0.2$ within sliding windows of 50 SNVs and advancing by 5 SNVs at the time. The obtained LD-pruned dataset was finally filtered for variants with minor allele frequency <0.01 and used to compute PCA utilizing the *smartpca* method implemented in the EIGENSOFT package (*Patterson et al., 2006*), as well as to run the ADMIXTURE algorithm version 1.3.0 (*Alexander et al., 2009*) by testing K = 2 to K = 12 population clusters. In detail, 25 replicates with different random seeds were run for each K and we retained only those presenting the highest

log-likelihood values. In addition, cross-validation errors were calculated for each *K* to identify the value that best fit the data. Both PCA and ADMIXTURE results were visualized with the R software version 4.0.5. ADMIXTURE results were visualized with the R software version 4.0.5.

## Detecting signatures of AI

To identify chromosomal regions showing signatures putatively ascribable to AI events, we submitted the phased WGS dataset to the *VolcanoFinder* pipeline, which relies on the analysis of the allele frequency spectrum of the population that is supposed to have experienced archaic introgression (*Setter et al., 2020*). This model considers three populations: recipient (i.e., the modern population), donor (i.e., the archaic population), and outgroup and assumes the occurrence of the introgression event after which a beneficial haplotype is introduced in the modern gene pool and starts to rise in frequency because of demographic random processes and because of the action of natural selection on it. The influence of these evolutionary forces comports the existence of multiple haplotypes that carry the beneficial archaic allele, thus comporting elevated heterozygosity level tested by the LR statistic. Therefore, the pattern of variability tested by such a statistic deeply differs from that attributable to classical selective sweeps (especially from that associated to a hard selective sweep), allowing to identify weak signatures of adaptation that resulted from both the introgression and the action of natural selection on beneficial standing genetic variation.

The *VolcanoFinder* algorithm was chosen among several approaches developed to detect AI signatures according to the following reasons. First of all, it can test jointly both archaic introgression and adaptive evolution according to a model that differs from those considered by other statistics that are aimed at identifying chromosomal segments showing low divergence with respect to a specific archaic sequence and/or enriched in alleles uniquely shared between the admixed group and the archaic source and characterized by a frequency above a certain threshold in the population under study (*Racimo et al., 2017*). In fact, these methods are especially useful to test an evolutionary scenario conformed to that expected in the case that adaptation was mediated by hard selective sweeps. On the contrary, *VocanoFinder* was proved to have an elevated power in the identification of AI events mediated by more than one predominant haplotype (*Setter et al., 2020*), as expected when soft sweeps/polygenic adaptations occurred. Moreover, *VolcanoFinder* relies on less demanding computational efforts with respect to algorithms that require to be trained on genomic simulations built specifically to reflect the evolutionary history of the population under study (*Gower et al., 2021*; *Zhang et al., 2023*), but possibly introducing bias in the obtained results if the information that guides simulations is not accurate.

We thus scanned Tibetan WGS data using the *VolcanoFinder* method and calculating two statistics for each polymorphic site: $\alpha$ (subsequently converted in $-\log\alpha$) and LR which are informative, respectively, of: (1) the strength of natural selection and (2) the conformity to the evolutionary model of AI. Since elevated LR scores are assumed to support AI signatures (*Setter et al., 2020*), we filtered the most significant results obtained by focusing on SNVs showing LR values falling in the positive tail (i.e., top 5%) of the distribution built for such a statistic. We then visualized the distribution of both $\alpha$ and LR parameters across the genomic regions including the *EPAS1* and *EGLN1* genes, which we considered, respectively, as positive and negative control loci for AI and by investigating chromosomal intervals spanning 50 kb up- and downstream to these genes. We then defined the new candidate AI genomic regions based on their conformity with the pattern observed for the positive AI control gene (i.e., according to the detection of multiple peaks of LR scores consistently distributed in the entire genomic region considered and coupled with elevated values of the $-\log\alpha$ parameter). Moreover, we relied on evidence advanced by previous studies aimed at detecting archaic introgression signatures from WGS data for individuals with Tibetan and/or Han Chinese ancestries (*Hu et al., 2017*; *Browning et al., 2018*; *Zhang et al., 2021*) to filter out loci potentially targeted by natural selection in Tibetans, but with questionable archaic origins.

## Identifying gene networks including Denisovan-like derived alleles

To validate archaic introgression signatures inferred with *VolcanoFinder* by using an independent method, we followed the *Signet* approach described by *Gouy and Excoffier, 2020*, with the aim of identifying biological functions whose underlying genomic regions might have been significantly shaped by Denisovan introgression. The *Signet* approach consists in crosschecking the information

contained in the input dataset with that collected in reference databases of functional pathways, such as the KEGG (available at https://www.kegg.jp/), using a simulated annealing algorithm approach to define the High Scoring Subnetworks within each biological pathway (*Gouy and Excoffier, 2020*). In detail, we used the *Signet* algorithm to reconstruct network of genes that participate to the same biological pathway and that also included Denisovan-like derived alleles observable in the Tibetan population but not in an outgroup population of African ancestry, by assuming that only Eurasian *H. sapiens* populations experienced Denisovan admixture. To do so, we first compared the Tibetan and Denisovan genomes to assess which SNVs were present in both modern and archaic sequences. These loci were further compared with the ancestral reconstructed refence human genome sequence to discard those presenting an ancestral state (i.e., that we have in common with several primate species). Moreover, we further filtered the considered variants by retaining only those alleles that were completely absent (i.e., with frequency equal to zero) in the YRI population sequenced by the 1000 Genomes Project (*Auton et al., 2015*). We then calculated the average frequency in the Tibetan population of the Denisovan-like derived alleles observable in each gene and we used the obtained genomic distribution to inform *Signet*. We performed five independent runs of the *Signet* algorithm to check for consistency of the significant gene networks and functional pathways identified and we finally depicted the confirmed results using Cytoscape v3.9.1 (*Shannon et al., 2003*).

## Testing adaptive evolution of candidate introgressed loci

To confirm signatures ascribable to the action of natural selection at the putative introgressed loci supported by both *VolcanoFinder* and *Signet* analyses, we applied the *LASSI* likelihood method (*Harris and DeGiorgio, 2020*) on the available Tibetan WGS data. Such an approach detects and distinguishes genomic regions that have experienced different types of selective events (i.e., strong and weak ones) and has been demonstrated to be more powerful in doing so than other haplotype-based approaches (*Harris and DeGiorgio, 2020*). The rationale behind the *LASSI* approach is based on the recognition of the modification resulted in the haplotype frequency spectrum of a given genomic region after the action of natural selection on it. For instance, according to the hard sweep model, when in the population arise a beneficial mutation with a very strong impact on a given phenotypic trait, the haplotype frequency spectrum of such a genomic region will be characterized by a single haplotype with an extremely elevated frequency in the population (i.e., the sweeping haplotype), while the other haplotypes whether they exist, are found at very low frequencies. Consequentially, when the selection acts on few new variants with a lower impact on a trait or on standing genetic variation, the resulted haplotype spectrum will be characterized by the existence of two or few more haplotypes that reach moderate frequencies in the population. Conversely, the variability pattern associated to the haplotype frequency spectrum expected under neutrality will be characterized by a series of different haplotypes at low frequencies in the population.

Specifically, we calculated for each genomic window the likelihood $T$ statistic, which measures the conformity of variability patterns of the analysed region to those expected according to a haplotype frequency spectrum under adaptive rather than neutral evolution. In addition, the *LASSI* algorithm calculated the parameter $m$ (i.e., the number of sweeping haplotypes) for each genomic region, thus classifying them as affected by hard sweeps (when $m = 1$) or soft sweeps (when $m > 1$). In detail, $T$ scores significantly different from zero indicate the conformity with a non-neutral evolutionary scenario, with ever higher likelihood scores being indicative of increasingly robust evidence for a selective event (*Harris and DeGiorgio, 2020*).

The method requires to fix a custom value for the length of the considered genomic windows, which are measured in terms of the number of SNVs included in them, and to move windows by 1 SNV after each computation. Therefore, we selected this fixed-length value (i.e., 13 SNVs) by estimating the average number of SNVs included into a haplotype block as defined for the population under study by using the `--blocks` function implemented in PLINK v1.9 (*Purcell et al., 2007*). Moreover, by following the indications by *Harris and DeGiorgio, 2020* of choosing values for the fixed number of haplotypes in the spectrum (i.e., $K$ values) <10 for increasing the power of the $T$ statistic, we set it at seven. Finally, we choose the likelihood model 3 to calculate the $T$ statistic and we applied the *LASSI* algorithm to the phased WGS dataset.

We then focused on the genomic windows showing $T$ scores falling in the positive tail (i.e., top 5%) of the obtained distribution and we crosschecked these results with those significant ones pointed out

by *VolcanoFinder* and *Signet* analyses to shortlist genomic regions having plausibly experienced both archaic introgression and adaptive evolution.

### *Haplostrip* analysis

To explicitly test whether the putative adaptive introgressed loci pointed out by *VolcanoFinder*, *Signet*, and *LASSI* analyses present variation patterns compatible with a scenario of introgression from the Denisovan archaic human species, we estimated genetic distance between modern and archaic haplotypes inferred for those genomic windows supported by all the methods mentioned above, as well as for *EPAS1* and *EGLN1* for the sake of comparison with established positive and negative control genes that have been previously proved to be involved or not in AI events. For this purpose, we used the *Haplostrip* pipeline, as described in previous studies (*Huerta-Sánchez et al., 2014*; *Marnetto and Huerta-Sánchez, 2017*). Moreover, since the *EGLN1* gene did not include any Denisovan-like variant as identified according to the criteria described in the previous paragraphs, we choose to build the *Haplostrips* heatmap by considering the *EGLN1* genomic window associated with the highest value of the *LASSI* statistic.

In detail, we merged the 27 Tibetan whole genomes under study (*Jeong et al., 2018*) with 27 CHB, 27 YRI WGSs (*Auton et al., 2015*) and with the Denisovan genome (*Meyer et al., 2013*) (downloadable at http://cdna.eva.mpg.de/neandertal/altai/Denisovan/). The CHB population, which is known to share a relatively recent common ancestry with Tibetans, was used as a 'negative low-altitude control' (i.e., as a group whose ancestors experienced Denisovan introgression, but did not evolve high-altitude adaptation). YRI individuals were instead used as the outgroup population (i.e., a population that presumably did not experience Denisovan admixture), as previously proposed (*Zhang et al., 2021*). We then phased the assembled dataset with SHAPEIT2 v2.r904 (*Delaneau et al., 2013*), as described in the *Dataset composition and curation* section and we run the *Haplostrip* algorithm.

## Acknowledgements

We acknowledge support from the Fondazione Cassa di Risparmio in Bologna through the project 'Genetic adaptation and acclimatization to high altitude as experimental models to investigate the biological mechanisms that regulate physiological responses to hypoxia', which was granted to MS (n. 2019.0552).

## Additional information

### Funding

| Funder | Grant reference number | Author |
| --- | --- | --- |
| Fondazione Cassa di Risparmio in Bologna | 2019.0552 | Marco Sazzini |

The funders had no role in study design, data collection, and interpretation, or the decision to submit the work for publication.

### Author contributions

Giulia Ferraretti, Paolo Abondio, Data curation, Software, Formal analysis, Investigation, Writing – original draft; Marta Alberti, Software, Formal analysis; Agnese Dezi, Phurba T Sherpa, Paolo Cocco, Massimiliano Tiriticco, Marco Di Marcello, Guido Alberto Gnecchi-Ruscone, Luca Natali, Angela Corcelli, Giorgio Marinelli, Davide Peluzzi, Data curation, Writing – review and editing; Stefania Sarno, Formal analysis, Writing – original draft; Marco Sazzini, Conceptualization, Resources, Supervision, Funding acquisition, Writing – review and editing

### Author ORCIDs

Marco Sazzini (ID) https://orcid.org/0000-0001-5382-7827

### Ethics

The University of Bologna Ethics Committee released approval (Prot. 205142, 12/9/2019) for the present study within the framework of the project 'Genetic adaptation and acclimatization to high altitude as experimental models to investigate the biological mechanisms that regulate physiological responses to hypoxia'. However, no new sampling campaign was performed in the context of the present study and all the genomic data analysed were publicly available. The informed consent for the 27 Tibetan WGS analysed here was previously obtained and declared in the Ethics statement section of the study by Jeong et al., 2018.

Reviewer #1 (Public review): https://doi.org/10.7554/eLife.89815.3.sa1
Reviewer #2 (Public review): https://doi.org/10.7554/eLife.89815.3.sa2
Author response https://doi.org/10.7554/eLife.89815.3.sa3

## Additional files

### Supplementary files

• Supplementary file 1. Supplementary tables 1a-1f. (**a**) Populations included in the extended dataset. (**b**) Single-nucleotide variants (SNVs) associated with values falling in top 5% of the distribution of likelihood ratio (LR) statistic calculated by *VolcanoFinder*. (**c**) SNVs associated with values falling in top 5% of the distribution of the LR statistic and comprised in the genomic region of the *EPAS1* gene (i.e., 2:46474546–2:46663836). (**d**) SNVs associated with values falling in top 5% of the distribution of the LR statistic and comprised in the genomic region of the *EGLN1* gene (i.e., 1:231449502–1:231608033). (**e**) Adaptive intregressed genes confirmed by *VolcanoFinder* and previous studies. (**f**) Gene networks including Denisovan-like derived alleles identified according to the *Signet* approach.

• MDAR checklist

### Data availability

The current manuscript is a computational study, so no data have been generated for this manuscript. The dataset used has been generated by *Jeong et al., 2018*. The code and the software used have been developed by *Marnetto and Huerta-Sánchez, 2017*; *Setter et al., 2020*; *Gouy and Excoffier, 2020*; *Harris and DeGiorgio, 2020*.

The following previously published dataset was used:

| Author(s) | Year | Dataset title | Dataset URL | Database and Identifier |
|---|---|---|---|---|
| Jeong C, Witonsky DB, Basnyat B, Neupane M, Beall CM, Childs G, Craig SR, Novembre J, Di Rienzo A | 2018 | Tibetan/Sherpa Sequence Reads | https://www.ncbi.nlm.nih.gov/bioproject/PRJNA420511/ | NCBI BioProject, PRJNA420511 |

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
