## [Editor Report · eLife Assessment]

This study presents **valuable** findings on what networks of genes were impacted by introgression from Denisovans, to identify the biological functions involved in high-altitude adaptation in Tibet. This study applies **solid** and previously validated methodology to identify genes with signatures of both introgression and positive selection. This paper would be of interest to population geneticists, anthropologists, and scientists studying the genetic basis underlying high-altitude adaptation.

---

## [Referee Report · Reviewer #1 (Public review)]

Over the last decade, numerous studies have identified adaptation signals in modern humans driven by genomic variants introgressed from archaic hominins such as Neanderthals and Denisovans. One of the most classic signals comes from a beneficial haplotype in the EPAS1 gene in Tibetans that is evidently of Denisovan origin and facilitated high altitude adaptation (HAA). Given that HAA is a complex trait with numerous underlying genetic contributions, in this paper Ferraretti et al. asked whether Denisovan introgression facilitated HAA in other ways by contributing to additional HAA-related genetic variants. Specifically, the authors considered that if such signature exists, they most likely are only mild signals from polygenic selection, or soft sweeps on standing archaic variation, in contrast to a strong and nearly complete selection signal like the EPAS1. They leveraged a few recently developed methods, including a composite likelihood method for detecting adaptive introgression and a biological network-based method for detecting polygenic selection, and identified compelling evidence of additional genes that exhibit Denisovan-like adaptive introgression signature and contributed to the polygenic adaptation at high altitude in Tibetans.

Strength:

The study is well motivated by an important question, which is, whether archaic introgression can drive polygenic adaptation via multiple small effect contributions in genes underlying different biological pathways regulating a complex trait (such as HAA). This is a valid question and the influence of archaic introgression on polygenic adaptation has not been thoroughly explored by previous studies

The authors reexamined previously published high-altitude Tibetan whole genome data and detected new evidence of adaptive introgression and polygenic selection. Specifically, by applying VolcanoFinder, they confirmed previously identified adaptive introgression alleles such as EPAS1 and PPARA. By applying signet, they identified subsets of biological pathways enriched for archaic variants that contributed to HAA polygenic selection. They also leveraged additional methods such as LASSI and haplotype plotting to help confirm the signature of natural selection on their newly discovered adaptive introgression candidate genes.

Weakness:

The manuscript also improved substantially since the initial review, and the new candidate genes presented here now harbor compelling and convincing evidence of both adaptive introgression and HAA polygenic selection. There are no notable weaknesses in the revised manuscript and updated results.

---

## [Referee Report · Reviewer #2 (Public review)]

Summary:

In Ferrareti et al. they identify adaptively introgressed genes using VolcanoFinder and then identify pathways enriched for adaptively introgressed genes. They use signet to identify pathways that are enriched for Denisovan alleles. The authors find that angiogenesis is one of the biological functions enriched for introgression.

Strengths:

Most papers that have studied the genetic basis of high altitude (HA) adaptation in Tibet have highly emphasized the role of a few genes (e.g. EPAS1, EGLN1), and in this paper the authors look for more subtle signals of selection in other genes to investigate how archaic introgression may be enriched at the pathway level. A couple of methods are used to confirm the consistency of the results.

Looking into the biological functions enriched for Denisovan introgression in Tibetans is important for characterizing the impact of Denisovan introgression in facilitating high altitude adaptations.

Weaknesses:

I thank the authors for providing an improved version of their manuscript.

---

## [Author Response]

The following is the authors’ response to the original reviews.

**Reviewer #1 (Public Review):**
Over the last decade, numerous studies have identified adaptation signals in modern humans driven by genomic variants introgressed from archaic hominins such as Neanderthals and Denisovans. One of the most classic signals comes from a beneficial haplotype in the *EPAS1* gene in Tibetans that is evidently of Denisovan origin and facilitated high altitude adaptation (HAA). Given that HAA is a complex trait with numerous underlying genetic contributions, in this paper Ferraretti et al. asked whether additional HAA-related genes may also exhibit a signature of adaptive introgression. Specifically, the authors considered that if such a signature exists, they most likely are only mild signals from polygenic selection, or soft sweeps on standing archaic variation, in contrast to a strong and nearly complete selection signal like in the *EPAS1*. Therefore, they leveraged two methods, including a composite likelihood method for detecting adaptive introgression and a biological networkbased method for detecting polygenic selection, and identified two additional genes that harbor plausible signatures of adaptive introgression for HAA.Strengths:The study is well motivated by an important question, which is, whether archaic introgression can drive polygenic adaptation via multiple small effect contributions in genes underlying different biological pathways regulating a complex trait (such as HAA). This is a valid question and the influence of archaic introgression on polygenic adaptation has not been thoroughly explored by previous studies.The authors reexamined previously published high-altitude Tibetan whole genome data and applied a couple of the recently developed methods for detecting adaptive introgression and polygenic selection.Weaknesses:My main concern with this paper is that I am not too convinced that the reported genomic regions putatively under polygenic selection are indeed of archaic origin. Other than some straightforward population structure characterizations, the authors mainly did two analyses with regard to the identification of adaptive introgression: First, they used one composite likelihood-based method, the VolcanoFinder, to detect the plausible archaic adaptive introgression and found two candidate genes (*EP300* and *NOS2*). Next, they attempted to validate the identified signal using another method that detects polygenic selection based on biological network enrichments for archaic variants.In general, I don't see in the manuscript that the choice of methods here are well justified. VolcanoFinder is one among the several commonly used methods for detecting adaptive introgression (eg. the D, RD, U, and Q statistics, genomatnn, maldapt etc.). Even if the selection was mild and incomplete, some of these other methods should be able to recapitulate and validate the results, which are currently missing in this paper. Besides, some of the recent papers that studied the distribution of archaic ancestry in Tibetans don't seem to report archaic segments in the two gene regions. These all together made me not sure about the presence of archaic introgression, in contrast to just selection on ancestral variation.Furthermore, the authors tried to validate the results by using signet, a method that detects enrichments of alleles under selection in a set of biological networks related to the trait. However, the authors did not provide sufficient description on how they defined archaic alleles when scoring the genes in the network. In fact, reading from the method description, they seemed to only have considered alleles shared between Tibetans and Denisovans, but not necessarily exclusively shared between them. If the alleles used for scoring the networks in Signet are also found in other populations such as Han Chinese or Africans, then that would make a substantial difference in the result, leading to potential false positives.Overall, given the evidence provided by this article, I am not sure they are adequate to suggest archaic adaptive introgression. I recommend additional analyses for the authors to consider for rigorously testing their hypothesis. Please see the details in my review to the authors.
**Reviewer #2 (Public Review):**
In Ferrareti et al. they identify adaptively introgressed genes using VolcanoFinder and then identify pathways enriched for adaptively introgressed genes. They also use a signet to identify pathways that are enriched for Denisovan alleles. The authors find that angiogenesis and nitric oxide induction are enriched for archaic introgression.Strengths:Most papers that have studied the genetic basis of high altitude (HA) adaptation in Tibet have highly emphasized the role of a few genes (e.g. *EPAS1*, *EGLN1*), and in this paper, the authors look for more subtle signals in other genes (e.g *EP300*, *NOS2*) to investigate how archaic introgression may be enriched at the pathway level.Looking into the biological functions enriched for Denisovan introgression in Tibetans is important for characterizing the impact of Denisovan introgression.Weaknesses:The manuscript lacks details or justification about how/why some of the analyses were performed. Below are some examples where the authors could provide additional details.The authors made specific choices in their window analysis. These choices are not justified or there is no comment as to how results might change if these choices were perturbed. For example, in the methods, the authors write "Then, the genome was divided into 200 kb windows with an overlap of 50 kb and for each of them we calculated the ratio between the number of significant SNVs and the total number of variants."Additional information is needed for clarity. For example, "we considered only protein-protein interactions showing confidence scores {greater than or equal to} 0.7 and the obtained protein frameworks were integrated using information available in the literature regarding the functional role of the related genes and their possible involvement in high-altitude adaptation." What do the confidence scores mean? Why 0.7?In the method section (Identifying gene networks enriched for Denisovan-like derived alleles), the authors write "To validate VolcanoFinder results by using an independent approach". Does this mean that for signet the authors do not use the regions identified as adaptively introgressed using volcanofinder? I thought in the original signet paper, the authors used a summary describing the amount of introgression of a given region.Later, the authors write "To do so, we first compared the Tibetan and Denisovan genomes to assess which SNVs were present in both modern and archaic sequences. These loci were further compared with the ancestral reconstructed reference human genome sequence (1000 Genomes Project Consortium et al., 2015) to discard those presenting an ancestral state (i.e., that we have in common with several primate species)." It is not clear why the authors are citing the 1000 genomes project. Are they comparing with the reference human genome reference or with all populations in the 1000 genomes project? Also, are the authors allowing derived alleles that are shared with Africans? Typically, populations from Africa are used as controls since the Denisovan introgression occurred in Eurasia.The methods section for Figures 4B, 4C, and 4D is a little hard to understand. What is the x-axis on these plots? Is it the number of pairwise differences to Denisovan? The caption is not clear here. The authors mention that "Conversely, for non-introgressed loci (e.g., *EGLN1*), we might expect a remarkably different pattern of haplotypes distribution, with almost all haplotype classes presenting a larger proportion of non-Tibetan haplotypes rather than Tibetan ones." There is clearly structure in *EGLN1*. There is a group of non-Tibetan haplotypes that are closer to Denisovan and a group of Tibetan haplotypes that are distant from Denisovan...How do the authors interpret this?In the original signet paper (Guoy and Excoffier 2017), they apply signet to data from Tibetans. Zhang et al. PNAS (2021) also applied it to Tibetans. It would be helpful to highlight how the approach here is different.

We thank the Reviewers for having appreciated the rationale of our study and to have identified potential issues that deserve to be addressed in order to better focus on robust results specifically supported by multiple approaches.

First, we agree with the Reviewers that clarification and justification for the methodologies adopted in the present study should be deepened with respect to what done in the original version of the manuscript, with the purpose of making it more intelligible for a broad range of scientists. As reported thoroughly in the revised version of the text, the VolcanoFinder algorithm, which we used as the primary method to discover new candidate genomic regions affected by events of adaptive introgression, was chosen among several approaches developed to detect signatures ascribable to such an evolutionary process according to the following reasons: (i) VolcanoFinder is one of the few methods that can test jointly events of both archaic introgression and adaptive evolution (e.g., the D statistic cannot formally test for the action of natural selection, having been also developed to provide genome wide estimates of allele sharing between archaic and modern groups rather than to identify specific genomic regions enriched for introgressed alleles); (ii) the model tested by the VolcanoFinder algorithm remarkably differs from those considered by other methods typically used to test for adaptive introgression, such as the RD, U and Q statistics, which are aimed at identifying chromosomal segments showing low divergence with respect to a specific archaic sequence and/or enriched in alleles uniquely shared between the admixed group and the source population, as well as characterized by a frequency above a certain threshold in the population under study, thus being useful especially to test an evolutionary scenario conformed to that expected in the case that adaptation was mediated by strong selective sweeps rather than weak polygenic mechanisms (see answer to comment #1 of Reviewer #1 for further details); (iii) VolcanoFinder relies on less demanding computational efforts respect to other algorithms, such as genomatnn and Maladapt, which also require to be trained on large genomic simulations built specifically to reflect the evolutionary history of the population under study, thus increasing the possibility to introduce bias in the obtained results if the information that guides simulation approaches is not accurate.

Despite that, we agree with Reviewer #2 that some criteria formerly implemented during the filtering of VolcanoFinder results (e.g., normalization of LR scores, use of a sliding windows approach, and implementation of enrichment analysis based on specific confidence scores) might introduce erratic changes, which depend on the thresholds adopted, in the list of the genomic regions considered as the most likely candidates to have experienced adaptive introgression. To avoid this issue, and to adhere more strictly to the VolcanoFinder pipeline of analyses developed by Setter et al. 2020, in the revised version of the manuscript we have opted to use raw LR scores and to shortlist the most significant results by focusing on loci showing values falling in the top 5% of the genomic distribution obtained for such a statistic (see Materials and methods for details).

Moreover, to further reduce the use of potential arbitrary filtering thresholds we decided to do not implement functional enrichment analysis to prioritize results from the VolcanoFinder method. To this end, although a STRING confidence score (i.e., the approximate probability that a predicted interaction exists between two proteins belonging to the same functional pathway according to information stored in the KEGG database) above 0.7 is generally considered a high confidence score (string-db.org, Szklarczyk et al. 2014), we replaced such a prioritization criterion by considering as the most robust candidates for adaptive introgression only those genomic regions that turned out to be supported by all the approaches used (i.e., VolcanoFinder, Signet, LASSI and Haplostrips analyses).

According to the Reviewers’ comments on the use of the Signet algorithm, we realized that the rationale beyond such a validation approach was not well described in the original version of the manuscript. First and foremost, we would like to clarify that in the present study we did not use this method to test for the action of natural selection (as it was formerly used by Gouy et al. 2017), but specifically to identify genomic regions putatively affected by archaic introgression. For this purpose, we followed the approach described by Gouy and Excoffier 2020 by searching for significant networks of genes presenting archaic-derived variants observable in the considered Tibetan populations but not in an outgroup population of African ancestry. Accordingly, we used the Signet method as an independent approach to obtain a first validation of introgressed (but not necessarily adaptive) loci pointed out by VolcanoFinder results.

In detail, in response to the question by Reviewer #2 about which genomic regions have been considered in the Signet analysis, it is necessary to clarify that to obtain the input score associated to each gene along the genome, as required by the algorithm, we calculated average frequency values per gene by considering all the archaic-derived alleles included in the Tibetan dataset but not in the outgroup one. Therefore, we did not take into account only those loci identified as significant by VolcanoFinder analysis, but we performed an independent genome scan. Then, we crosschecked significant results from VolcanoFinder and Signet approaches and we shortlisted the genomic regions supported by both. This approach thus differs from that of Zhang et al. 2021 in which the input scores per gene were obtained by considering only those loci previously pointed out by another method as putatively introgressed. Moreover, as mentioned in the previous paragraph, our approach differs also from that implemented by Guoy et al. 2017, in which the input scores assigned to each gene were represented by the variants showing the smallest P-value associated to a selection statistic, being thus informative about putative adaptive events but not introgression ones.

However, as correctly pointed out by both the Reviewers, we formerly performed Signet analysis by considering derived alleles shared between Tibetans and the Denisovan species, without filtering out those alleles that are observed also in other modern human populations. We agree with the Reviewers that this approach cannot rule out the possibility of retaining false positive results ascribable to ancestral polymorphisms rather than introgressed alleles. According to the Reviewers’ suggestion, we thus repeated the Signet analysis by removing derived alleles observed also in an outgroup population of African ancestry (i.e., Yoruba), by assuming that only Eurasian *H. sapiens* populations experienced Denisovan admixture. In detail, we considered only those alleles that: (i) were shared between Tibetans and Denisovan (i.e., Denisovan-like alleles); (ii) were assumed to be derived according to the comparison with the ancestral reconstructed reference human genome sequence; (iii) were completely absent (i.e., present frequency equal to zero) in the Yoruba population sequenced by the 1000 Genomes Project. Despite the comment of Reviewer #1 seems to propose the possible use of Han Chinese as a further control population, we decided to do not filter out Denisovan-like derived alleles present also in this human group because evidence collected so far suggest that Denisovan introgression in the gene pool of East Asian ancestors predated the split between low-altitude and high-altitude populations (Lu et al. 2016; Hu et al. 2017) and, as mentioned before, we aimed at using the Signet algorithm to validate introgression events rather than adaptive ones (see the answer to comment #6 of Reviewer #1 for further details). Moreover, we would like to remark that we decided to maintain the Signet analysis as a validation method in the revised version of the manuscript because: (i) comments from both the Reviewers converge in suggesting how to effectively improve this approach, and (ii) it represents a method that goes beyond the simple identification of single putative introgressed alleles, by instead enabling us to point out those biological functions that might have been collectively shaped by gene flow from Denisovans.

In addition to validate genomic regions putatively affected by archaic introgression by crosschecking results from the VolcanoFinder and Signet analyses, according to the suggestion by Reviewer #1 we implemented a further validation procedure aimed at formally testing for the adaptive evolution of the identified candidate introgressed loci. For this purpose, we applied the LASSI likelihood haplotype based method (Harris & DeGiorgio 2020) to Tibetan whole genome data. Notably, we choose this approach mainly for the following reasons: (i) because it is able to detect and distinguish genomic regions that have experienced different types of selective events (i.e. strong and weak ones); (ii) it has been demonstrated to have increased power in identifying them with respect to other selection statistics (e.g., H12 and nSL) (Harris & DeGiorgio 2020). Again, we performed an independent genome scan using the LASSI algorithm and then we crosschecked the obtained significant results with those previously supported by VolcanoFinder and Signet approaches in order to shortlist genomic regions that have plausibly experienced both archaic introgression and adaptive evolution.

Moreover, we maintained a final validation step represented by Haplostrips analysis, which was instead specifically performed on chromosomal segments supported by results from both VolcanoFinder, Signet, and LASSI approaches. This enabled us to assess the similarity between Denisovan haplotypes and those observed in Tibetans (i.e., the population under study in which archaic alleles might have played an adaptive role in response to high-altitude selective pressures), Han Chinese (i.e., a sister group whose common ancestors with Tibetans have experienced Denisovan admixture, but have then evolved at low altitude), and Yoruba (i.e., an outgroup that is assumed to have not received gene flow from Denisovans).

In conclusion, we believe that the substantial changes incorporated in the manuscript according to the Reviewers’ suggestions strongly improved the study by enabling us to focus on more solid results with respect to those formerly presented. Interestingly, although the single candidate loci supported by all the approaches now implemented for validating the obtained results have attained higher prioritization with respect to previous ones (which are supported by some but not all the adopted methods), angiogenesis still stands out as the one of the main biological functions that have been shaped by events of adaptive introgression in human groups of Tibetan ancestry. This provides new evidence for the contribution of introgressed Denisovan alleles other than the *EPAS1* ones in modulating the complex adaptive responses evolved by Himalayan populations to cope with selective pressures imposed by high altitudes.

**Responses to Recommendations For The Authors:**

**Reviewer #1:**
The authors mainly relied on one method, VolcanoFinder (VF), to detect adaptive introgression signals. As one of the recently developed methods, VF indeed demonstrated statistical power at detecting mild selection on archaic variants, as well as detecting soft sweeps on standing variations. However, compared to other commonly used methods for detecting adaptive introgression, such as the U and Q stats (Racimo et al. 2017), genomatnn (Gower et al. 2021), or MaLAdapt (Zhang et al. 2023),VF doesn't seem to have better power at capturing mild and incomplete sweeps. And it makes me wonder about the justification for choosing VF over other methods here, which is not clearly explained in the manuscript. If these adaptive introgression candidates are legitimate, even if the signals are mild, at least some of the other methods should be able to recapitulate the signature (even if they don't necessarily make it through the genome-wide significance thresholds). I would be more convinced about the archaic origin of these regions if the authors could validate their reported findings using some of the aforementioned other methods.

According to the Reviewer’s suggestion, in the revised version of the manuscript we have expanded the considerations reported as concern the rationale that guided the choice of the adopted methods. In particular, in the Materials and methods section (see page 12) we have specificed the reasons for having used the VolcanoFinder algorithm.

First, it represents one of the few approaches that relies on a model able to test jointly the occurrence of archaic introgression and the adaptive evolution of the genomic regions affected by archaic gene flow, without the need for considering the putative source of introgression. This was a relevant aspect for us, beacuse we planned to adopt at least two main independent (and possibly quite different in terms of the underlying approaches) methods to validate the identified candidate intregressed loci and the other algorithm we used (i.e., Signet) was explicitly based on the comparison of modern data with the archaic sequence. Accordingly, the model tested by VolcanoFinder differs from those considered by the RD, U and Q statistics. In fact, RD statistic is aimed at identifying regions of the genome with low divergence with respect to a given archaic reference, while the U/Q statistics can detect those chromosomal segments enriched in alleles that are (i) uniquely shared between the admixed group (e.g., Tibetans) and the source population (e.g., Denisovans), and (ii) that present a frequency above a specific threshold in the admixed population (Racimo et al. 2016). For instance, all the loci considered as likely involved in adaptive introgression events by Racimo et al. 2016 presented remarkable frequencies, with most of them showing values above 50%. That being so, we decided to do not implement these methods because we believe that they are more suitable for the detection of adaptive introgression events involving few variants with a strong effect on the phenotype, which comport a substantial increase in frequency in the population subjected to the selective pressure (i.e., cases such as that of *EPAS1*), while it appears challenging to choose an arbitrary frequency threshold appropriate for the detection of weak and/or polygenic selective events.

As regards the possible use of Maladapt or genomatnn approaches as validation methods, we believe that they rely on more demanding computational efforts with respect to the Signet algorithm and, above all, they have the disadvantage of requiring to be trained on simulated genomic data. This makes them more prone to the potential bias introduced in the obtained results by simulations that do not carefully reflect the evolutionary history of the population under study.

Overall, we do not agree with the Reviwer’s statement about the fact that we mainly relied on a single method to detect adaptive introgression signals because, as mentioned above, the Signet algorithm was specifically used to identify genomic regions putatively affected by introgression. This method relies on assumptions very similar to those described above for the U/Q statistics (e.g. it considers alleles uniquely shared between Tibetans and Denisovans), but avoids the necessity to select a frequency threshold to shortlist the most likely adaptive intregressed loci. In addition, according to another suggestion by the Reviewer we have now implemented a further approach to provide evidence for the adaptive evolution of the candidate introgressed loci (see response to comment #3).

As regards the use of Signet, based on comments from both the Reviewers we realized that the rationale beyond such a validation approach was not well described in the original version of the manuscript. First and foremost, we would like to clarify that in the present study we did not use this method to test for the action of natural selection (as it was formerly used by Gouy et al. 2017), but specifically to identify genomic regions putatively affected by archaic introgression. For this purpose, we followed the approach described by Gouy and Excoffier (2020) by searching for significant networks of genes presenting archaic-derived variants observable in the considered Tibetan populations. That being so, we used the Signet method as an independent approach to obtain a first validation of VolcanoFinder results. However, by following suggestions from both the Reviweres, we modified the criteria adopted to filter for archaic-derived variants, by excluding those alleles in common between Denisovan and the Yoruba outgroup population (see response to comment #6 for further information regarding this aspect).

To sum up, we think that the combination of VolcanoFinder and Signet+LASSI approaches offered a good compromise between required computational efforts to shortlist the most robust candidates of adaptive introgressed loci and the typologies of model tested (i.e. that does not diascard *a priori* genomic signatures ascribable to weak and/or polygenic selective events). Morevoer, we would like to remark that we decided to maintain the Signet method as a validation approach in the revised version of the manuscript because: (i) comments from both the Reviewers converge in suggesting how to effectively improve this approach, and (ii) it represents a method that can be used to perform both single-locus validation analysis and to search for those biological functions that have been collectively much more impacted by archaic introgression, allowing to test a more realistic approximation of the polygenic model of adaptation involving introgressed alleles. In fact, although the single candidate loci supported by all the approaches now implemented for validating the obtained results (see responses to comments #3 and #7 for further details) have attained higher prioritization with respect to previous ones (i.e., *EP300* and *NOS2*, which are now supported by some but not all the adopted methods), angiogenesis still stands out as one of the main biological functions that have been shaped by events of adaptive introgression in the ancestors of Tibetan populations.

Besides, I am a little surprised to see that in Supplementary Figure 2, VF didn't seem to capture more significant LR values in the *EPAS1* region (positive control of adaptive introgression) than in the negative control *EGLN1* region. The author explained this as the selection on *EPAS1* region is "not soft enough", which I find a bit confusing. If there is no major difference in significant values between the positive and negative controls, how would the authors be convinced the significant values they detected in their two genes are true positives? I would like to see more discussion and justification of the VF results and interpretations.

In the light of such a Reviewer’s observation and according to the Reviewer #2 overall comment on the procedures implemented for filtering VolcanoFinder results, we realized that both normalization of LR scores and the use of a sliding windows approach might introduce erratic changes, which depend on the thresholds adopted, in the list of the genomic regions considered as the most likely candidates to have experienced adaptive introgression. To avoid this issue, and to adhere more strictly to the VolcanoFinder pipeline of analyses developed by Setter et al. 2020, in the revised version of the manuscript we have opted to use raw LR scores and to shortlist the most significant results by focusing on loci showing values falling in the top 5% of the genomic distribution obtained for such a statistic (see Materials and methods, page 13 lines 4 -16 for further details).

By following this approach, we indeed observed a pattern clearer than that previously described, in which the distribution of LR scores in the *EPAS1* genomic region is remarkably different with respect to that obtained for the *EGLN1* gene (Figure 2 – figure supplement 1). More in detail, we identified a total of 19 *EPAS1* variants showing scores within the top 5% of LR values, in contrast to only three *EGLN1* SNVs. Moreover, LR values were collectively more aggregated in the *EPAS1* genomic region and showed a higher average value with respect to what observed for *EGLN1*. We reported LR values, as well as -log (a) scores calculated for these control genes in Supplement tables 3 and 4.

Nevertheless, we agree with the Reviewer that results pointed out by VolcanoFinder require to be confirmed by additional methods, which is was what we have done to define both new candidate adaptive intregressed loci and the considered positive/negative controls. In fact, validation analyses performed to confirm signatures of both archaic introgression and adaptive evolution (i.e., *Signet*, *LASSI* and Haplostrips) converged in indicating that Tibetan variability at the *EGLN1* gene does not seem to have been shaped by archaic introgression events but only by the action of natural selection (see Results, page 5 lines 3-9, page 6 lines 23-25, page 7 lines 29-36; Discussion page 14 lines 33-36; Figure 2 – figure supplement 1B and Figure 4 – figure supplement 1B, 3B and 3D), also according to what was previously proposed (Hu et al., 2017). On the other hand, results from all validation analyses confirmed adaptive introgression signatures at the *EPAS1* genomic region (see Results page 4 lines 32-37, page 5 lines 1-2 and 30-34, page 6 lines 23-29; Figure 3A, 3B and Figure 4 – figure supplement 1A, 3A and 3C).

Finally, as already reported in the former version of the manuscript, our choice of considering *EPAS1* and *EGLN1* respectively as positive and negative controls for adaptive introgression was guided by previous evidence suggesting these loci as targets of natural selection in high-altitude Himalayan populations (Yang et al., 2017; Liu et al., 2022), although only *EPAS1* was proved to have been involved also in an adaptive introgression event (Huerta-Sanchez et al., 2014; Hu et al., 2017).

With that being said, I suggest the authors try to first validate the signal of positive selection in the two gene regions using methods such as H2/H1 (Garud et al. 2015), iHS (Voight et al. 2006) etc. that have demonstrated power and success at detecting mild sweeps and soft sweeps, regardless of if these are adaptive introgression.

According to the Reviewer’s suggestion, we validated the new candidate adaptive introgressed loci by using also a method to formally test for the action of natural selection. In particular, we decided to use the *LASSI* (Likelihood-based Approach for Selective Sweep Inference) algorithm developed by Harris & DeGiorgio (2020) mainly for the following reasons: (i) it is able to identify both strong and weak genomic signatures of positive selection similarly to others approaches, but additionally it can distinguish these signals by explicitly classifying genomic windows affected by hard or soft selective sweeps; (ii) when applied on simulated data generated under different demographic models and by setting a range of different values for the parameters that describe a selective event (e.g., the time at which the beneficial mutation arose, the selection coefficient *s*) it has been proved to have an increased power with respect to traditional selection scans, such as nSL, H2/H1 and H12 (see Harris & DeGiorgio 2020 for further details).

According to such an approach, we were able to recapitulate signatures of natural selection previously observed in Tibetans for both *EPAS1* and *EGLN1* (Figure 4 – figure supplement 1 and 3C – 3D). We also obtained comparable patterns for our previous candidate adaptive introgressed loci (i.e., *EP300* and *NOS2*), as well as for the new ones that have been instead prioritized in the revised version of the manuscript according to consistent results also from VolcanoFinder, Signet and Haplostrips analyses (see Results, page 6 lines 30-35; Figure 4C, 4D, Figure 4 – figure supplement 2C and 2D).

With regard to the plausible archaic origin of the haplotypes under selection in these gene regions, my concern comes from the fact that other recent studies characterizing the archaic ancestry landscape in Tibetans and East Asians (eg. SPrime reports from Browning et al. 2018, as well as ArchaicSeeker reports from Yuan et al. 2021) didn't report archaic segments in regions overlapping with *EP300* and *NOS2*. So how would the authors explain the discrepancy here, that adaptive introgression is detected yet there is little evidence of archaic segments in the regions?

We thank the Reviewer for the comment and the references provided. However, we read the suggested articles and in both of them it does not seem that genomes from individuals of Tibetan ancestry have been analysed. Moreover, in the study by Yuan et al. 2021 we were not able to find any table or supplementary table reporting the genomic segments showing signatures of Denisovan-like introgression in East Asian groups, with only findings from enrichment analyses performed on significant results being described for the Papuan population. Anyway, as reported below in the response to comment #5, in line with what observed by the Reviwer as concerns the original version of the manuscript, according to the additional validation analyses implemented during this revison *EP300* and *NOS2* received lower prioritization with respect to other loci showing more robust signatures supporting introgression of Denisovan alleles in the gene pool of Tibetan ancestors (i.e., *TBC1D1*, *PRKAG2*, *KRAS* and *RASGRF2*). Three out of four of these genes are in accordance also with previously published results supporting introgression of Denisovan alleles in the ancestors of present-day Han Chinese (Browning et al. 2018) or directly in the Tibetan genomes (Hu et al. 2017) (see Results, page 5 lines 10-21 and Supplement table 5). Despite that, the reason why not all the candidate adaptive introgression regions detected by our analyses are found among results from Browning et al. 2018 can be represented by the fact that in Han Chinese this archaic variation could have evolved neutrally after the introgression events, thus preventing the identification of chromosomal segments enriched in putative archaic introgressed variants according to VolcanoFinder and LASSI approaches (which consider also the impact of natural selection). In fact, the Sprime method implemented by Browning et al. 2018 focuses only on introgression events rather than adaptive introgression ones. For instance, the Denisovan-like regions identified with Sprime in Han Chinese by such a study do not comprise at all the *EPAS1* region.

Additionally, looking at Figure 4 and Supplementary Figure 4, the authors showed haplotype comparisons between Tibetans, Denisovan, and Han Chinese for *EP300* and *NOS2* regions. However, in both figures, there are about equal number of Tibetans and Han Chinese that harbor the haplotype with somewhat close distance to the Denisovan genotype. And this closest haplotype is not even that similar to the Denisovan. So how would the authors rule out the possibility that instead of adaptive introgression, the selection was acting on just an ancestral modern human haplotype?

We agree with the Reviewer that according to the analyses presented in the original version of the manuscript haplotype patterns observed at *EP300* and *NOS2* loci by means of the Haplostrips approach cannot ruled out the possibility that their adaptative evolution involved ancestral modern human haplotypes. In fact, after the modifications implemented in the adopted pipeline of analyses based on the Reviewers’ suggestions, their role in modulating complex adaptations to high-altitudes was confirmed also by results obtained with the LASSI algorithm (in addition to results from previous studies Bigham et al., 2010; Zheng et al., 2017; Deng et al., 2019; X. Zhang et al., 2020), but their putative archaic origin received lower prioritization with respect to other loci, being not confirmed by all the analyses performed.

Furthermore, I have a question about how exactly the authors scored the genes in their network analysis using Signet. The manuscript mentioned they were looking for enrichment of archaic-like derived alleles, and in the methods section, they mentioned they used SNPs that are present in both Denisovan and Tibetan genomes but are not in the chimp ancestral allele state. But are these "derived" alleles also present in Han Chinese or Africans? If so, what are the frequencies? And if the authors didn't use derived alleles exclusively shared between Tibetans and Denisovans, that may lead to false positives of the enrichment analysis, as the result would not be able to rule out the selection on ancestral modern human variation.

As mentioned in the response to comment #1, by following the suggestions of both the Reviewers we have modified the criteria adopted for filtering archaic derived variants exclusively shared between Denisovans and Tibetans. In particular, we retained as input for Signet analysis only those alleles that (i) were shared between Tibetans and Denisovan (i.e., Denisovan-like alleles) (ii) were in their derived state and (iii) were completely absent (i.e., show frequency equal to zero) in the Yoruba population sequenced by the 1000 Genome Project and used here as an outgroup by assuming that only Eurasian *H. sapiens* populations experienced Denisovan admixture. We instead decided to do not filter out potential Denisovan-like derived alleles present also in the Han Chinese population because multiple evidence agreed at indicating that gene flow from Denisovans occurred in the ancestral East Asian gene pool no sooner than 48–46 thousand years ago (Teixeira et al. 2019; Zhang et al. 2021; Yuan et al. 2021), thus predating the split between low-altitude and high-altitude groups, which occurred approximately 15 thousand years ago (Lu et al. 2016; Hu et al. 2017). In fact, traces of such an archaic gene-flow are still detectable in the genomes of several low-altitude populations of East Asian ancestry (Yuan et al. 2021).

Concerning the above, I would also suggest the authors replot their Figure 4 and Figure S4 by adding the African population (eg. YRI) in the plot, and examine the genetic distance among the modern human haplotypes, in contrast to their distance to Denisovan.

According to the Reviewer’s suggestion, after having identified new candidate adaptive introgressed loci according to the revised pipeline of analyses, we run the Haplostrips algorithm by including in the dataset 27 individuals (i.e., 54 haplotypes) from the Yoruba population sequenced by the 1000 Genomes Project (Figure 4A, 4B, Figure 4 - figure supplement 2A, 2B, 3A).

**Reviewer #2:**
In the methods the authors write "Since composite likelihood statistics are not associated with pvalues, we implemented multiple procedures to filter SNVs according to the significance of their LR values." What does significance mean here?

After modifications applied to the adopted pipeline of analyses according to the Reviewers’ suggestions (see responses to public reviews and to comments #1, #3, #6, #7 of Reviewer #1), new candidate adaptive introgressed loci have been identified specifically by focusing on variants showing LR values falling in the top 5% of the genomic distribution obtained for such a statistic in order to adhere more strictly to the VolcanoFinder approach developed by Setter et al. 2020. Therefore, the related sentence in the materials and methods section was modified accordingly.

Signet should be cited the first time it appears in the manuscript. The citation in the references is wrong. It lists R. Nielsen as the last author, but R. Nielsen is not an author of this paper.

We thank the Reviewer for the comment. We have now mentioned the article by Gouy and Excoffier (2020) in the Results section where the Signet algorithm was first described and we have corrected the related reference.

I could not find Figure 5 which is cited in the methods in the main text. I assume the authors mean Supplementary Figure 5, but the supplementary files have Figure 4.

We thank the Reviewer for the comment. We have checked and modified figures included in the article and in the supplementary files to fix this issue.

I didn't see a table with the genes identified as adaptatively introgressed with VolcanoFinder. This would be useful as I believe this is the first time VolcanoFinder is being used on Tibetan data?

According to the Reviewer suggestion, we have reported in Supplement table 2 all the variants showing LR scores falling in the top 5% of the genomic distribution obtained for such a statistic, along with the associated α parameters computed by the VolcanoFinder algorithm.

It is easier for the reviewer if lines have numbers.

According to the Reviewer suggestion, we have included line numbers in the revised version of the manuscript.